# MINT: Evaluating LLMs in Multi-turn Interaction with Tools and Language Feedback

**Xingyao Wang**[1,*] **Zihan Wang**[1,2,*,†] **, Jiateng Liu**[1] **, Yangyi Chen**[1] **, Lifan Yuan**[1,†] **, Hao Peng**[1] **, Heng Ji**[1]

[1] University of Illinois Urbana-Champaign, [2] Renmin University of China

[1]{xingyao6,zihanw,jiateng5,yangyic3,haopeng,hengji}@illinois.edu

## Abstract

To solve complex tasks, large language models (LLMs) often require multiple rounds of interactions with the user, sometimes assisted by external tools. However, current evaluation protocols often emphasize benchmark performance with single-turn exchanges, neglecting the nuanced interactions among the user, LLMs, and external tools, while also underestimating the importance of natural language feedback from users. These oversights contribute to discrepancies between research benchmark evaluations and real-world use cases. We introduce **MINT**, a benchmark that evaluates LLMs' ability to solve challenging tasks with **m**ulti-turn **int**eractions by (1) using tools and (2) leveraging natural language feedback. To ensure reproducibility, we provide an evaluation framework where LLMs can access tools by executing Python code and receive users' natural language feedback simulated by GPT-4. We repurpose a diverse set of established evaluation datasets focusing on reasoning, coding, and decision-making and carefully curate them into a compact subset for efficient evaluation. Our analysis of 20 open- and closed-source LLMs offers intriguing findings. (a) LLMs generally benefit from tools and language feedback, with performance gains (absolute, same below) of 1–8% for each turn of tool use and 2–17% with natural language feedback. (b) Better single-turn performance does not guarantee better multi-turn performance. (c) Surprisingly, among the evaluated LLMs, supervised instruction-finetuning (SIFT) and reinforcement learning from human feedback (RLHF) generally hurt multi-turn capabilities. We expect MINT can help measure progress and incentivize research in improving LLMs' capabilities in multi-turn interactions, especially for open-source communities where multi-turn human evaluation can be less accessible compared to commercial LLMs with a larger user base. [1]

## 1 Introduction

To address complex tasks, Large Language Models (LLMs) often need multiple rounds of interaction with the users, sometimes aided by external tools (Schick et al., 2023b; ChatGPT Plugins; Mialon et al., 2023). LLMs' performance during multiple turns of user-LLM exchanges is crucial in real-world applications: roughly 73% of Human-ChatGPT conversations contain more than one turn based on 94k entries of ShareGPT data (2023). Meanwhile, the ability to adapt to user-provided natural language feedback is also pivotal for their practical utility. However, current LLM evaluations predominantly focus on *single-turn* input-output (Hendrycks et al., 2020; Chen et al., 2021) and often overlook user-provided natural language feedback (Liu et al., 2023d; Deng et al., 2023b; Yang et al., 2023a; Shridhar et al., 2020), creating a discrepancy between real-world use cases and evaluation. Measuring how much LLMs can benefit from **both** tools and natural language feedback during **multi-turn interaction** is essential to incentivize future research to improve LLMs' capabilities in a broader range of real-world scenarios.

To bridge these gaps, we introduce MINT. It is a benchmark for LLMs that measures their performance during **multi-turn interaction**, focusing on two particular capabilities (§2.1): (1) **tool-**

---

*Equal contribution. †Work done during internship at UIUC.

[1]Code is available on our project website: https://xingyaoww.github.io/mint-bench

**augmented task-solving**; (2) **leveraging natural language feedback**. MINT mirrors the real-world user-LLM-tool collaborative problem-solving setting. To solve a problem, the LLM can use external tools by generating and executing Python programs (Wang et al., 2024) and/or collecting natural language feedback to refine its solutions; the feedback is provided by GPT-4 (OpenAI, 2023), aiming to simulate human users in a reproducible and scalable way.[2] For a comprehensive evaluation, we include eight established datasets spanning reasoning, code generation, and decision-making (§2.2). To facilitate affordable multi-turn evaluation, after collecting 29,307 diverse instances from existing datasets (Tab. 1), we construct a subset of 586 challenging and representative instances that require multi-turn interaction to solve[3].

We evaluate 4 closed- and 16 open-source LLMs with MINT. We measure LLMs' **tool-augmented task-solving capability** by analyzing their performance from multi-turn tool use (§3.2). To assess the **ability to leverage natural language feedback**, we measure their performance upon natural language feedback by GPT-4 (§3.3). Our results show that:

- All models benefit from tool interaction and natural language feedback, with absolute performance gains by 1–8% for each additional turn of tool use, and 2–17% with natural language feedback.
- Better single-turn performance does *not* necessarily entail better multi-turn performance. For example, while Claude-2 outperforms its predecessor Claude-1 in single-turn evaluation, the latter benefit more from interaction and performs better with $> 2$ turns.
- There is a notable gap between open- and closed-source LLMs in multi-turn interaction performance. For example, with the help of language feedback, even the best open-source model, Lemur-70b-chat-v1, lags behind the best closed-source model by 8.7% in absolute success rate.
- On most LLMs we evaluated, models trained with supervised instruction fine-tuning (SIFT, Wei et al., 2022) and reinforcement learning from human feedback (RLHF, Ouyang et al., 2022a) perform worse in multi-turn settings regardless of the presence of language feedback. For example, SIFT hurts Codellama-34B's multi-turn performance by 11.1% and 15.4% (w/ feedback), and RLHF negatively affects LLaMA-2-70B by 8.5% and 8.7%, respectively. Notable exceptions are Vicuna-7B and Lemur-70b-chat-v1, where SIFT improves multi-turn interaction.

By fixing the LLM to evaluate and changing the feedback-provider LLM, MINT can measure different LLMs' capabilities in *providing useful feedback* (§3.4); We find that feedback-providing ability could be orthogonal to task-solving ability: despite performing the worst in task-solving, CodeLLaMA-34B-Instruct can provide feedback to improve the stronger GPT-3.5. Additionally, MINT's challenging evaluation reveals undesired artifacts in ShareGPT data (2023), a widely used dataset for instruction tuning (§3.5). Furthermore, we show that GPT4-simulated language feedback is as helpful as human-written feedback based on human evaluation and task performance (§3.6).

We expect that MINT can help track progress and incentivize future research in improving LLM's multi-turn task-solving and/or feedback-providing capabilities, especially for open-source communities where human evaluation can be less accessible than commercial LLMs with a large user base.

## 2 🌿 MINT

In this section, we discuss (1) how to evaluate multi-turn interaction (§2.1) with tool use and language feedback under different settings; (2) how we repurpose existing datasets for MINT evaluation (§2.2). We use Fig. 1 as a running example.

### 2.1 INTERACTION FRAMEWORK

MINT aims to simulate real-world applications of LLMs, emphasizing user-LLM and LLM-tool interactions. In user-LLM collaborative problem-solving, a human user provides initial instruction and aims to obtain a correct solution with little effort to help the LLM. On the other hand, augmenting LLMs' with tools can effectively improve LLMs' task-solving capabilities (Mialon et al., 2023), suggesting the importance of LLM-tool interaction. We instruct the LLM (§F.4.1) to perform the

---

[2]We use `gpt-4-0613` version in this work.

[3]Evaluating an LLM using MINT costs around 100 USD ($\approx$ 3M prompt tokens and $\approx$ 100K completion tokens) with feedback from `gpt-4-0613` ($0.03/1K prompt tokens and $0.06/1K completion tokens), roughly 7% of the cost compared to hiring real-human annotators (§3.6).

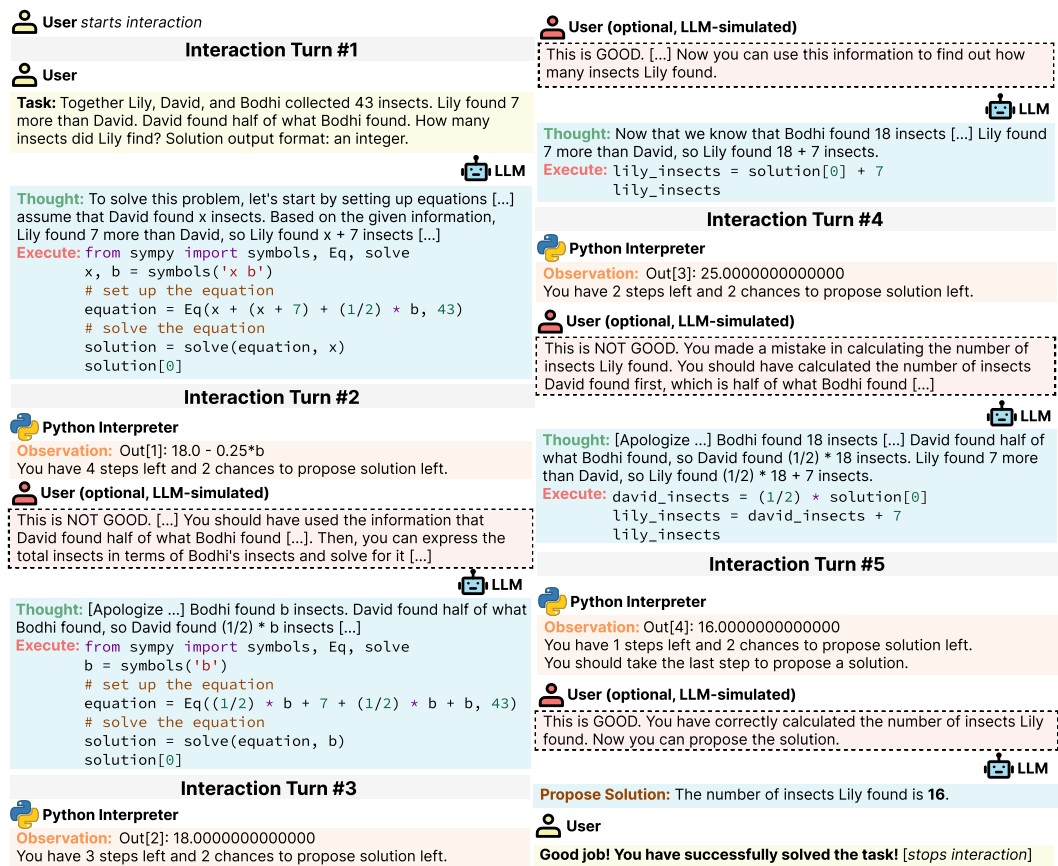

Figure 1: An interaction trajectory produced by evaluating `gpt-3.5-turbo-0613` with MINT on a mathematical reasoning task. The evaluated model's outputs are in the blue boxes, and the feedback by `gpt-4-0613` in red, dotted ones. Some details are omitted for clarity.

following steps *in each turn*: **(1)** optionally express its reasoning process ("Thought:" in Fig. 1, similar to Yao et al. 2022); **(2)** then either interact with tools by generating Python code and executing it through a Python interpreter ("Execute:" in Fig. 1), or proposing a solution to the user ("Propose Solution:" in Fig. 1). We adopt code as a unified tool interface due to its flexibility and performance, as demonstrated by Wang et al. (2024). In our implementation, the model is instructed to wrap their "Execute" and "Propose Solution" actions with pairs of `<execute>` and `<solution>` tags for ease of parsing. We standardize the prompts and in-context examples for different LLM variants (base vs. chat) and for task-solving and feedback providing, aiming for fair and reproducible comparisons (Appendix §F.4.1, §F.4.2, and §F.5). In what follows, we introduce three settings with increased interaction complexities to measure different aspects of multi-turn interaction.

**LLMs interacting with a lazy user.** We consider the scenario where a user provides an initial instruction and makes *minimal* efforts to guide the LLM toward the final solution. This setting covers real-world problems that are "hard to solve directly, but easy to verify": the user may not know how to solve the problem or understand the reasoning process, but it is still possible for the user to provide minimal feedback on the final outcome (e.g., "this is not what I want"). This will serve as a baseline for evaluating LLM's ability to solve tool-augmented tasks and leverage natural language feedback. The LLM is given *two* attempts to propose solutions for each problem, with a limit on the number of interaction turns $k$ (§3.1). Upon a proposed solution, MINT simulates users that check the solution's correctness with ground truths. When the first attempt is wrong, the user responds "Your answer is wrong." The interaction ends either after the LLM has made two attempts to propose a solution, or when a proposed solution is correct (5th turn of Fig. 1), or when the $k$-th turn of interaction is reached. We consider this as the case of *Lazy User-LLM Interaction* since the simulated user provides *at most one* additional binary feedback during interaction. We follow standard evaluation practice and use established evaluation metrics for each task in §2.2.

**LLMs interacting with a lazy user and tools.** Under the lazy User-LLM interaction setting, we measure the LLM's ability to solve tasks using tools by comparing their task-solving success rate

Table 1: Dataset statistics of re-purposed data instances from existing datasets into MINT. We filter and down-sample existing datasets to construct a compact set of complex tasks that require multi-turn interaction to solve (§2.2).

| Task Type | Task Name | Original Size | Reduced Size in MINT |
|---|---|---|---|
| **Code Generation** | HumanEval (Chen et al., 2021) | 164 | 45 |
| | MBPP (Austin et al., 2021) | 500 | 91 |
| **Decision Making** | ALFWorld (Shridhar et al., 2020) | 134 | 134 |
| **Reasoning** | GSM8K (Cobbe et al., 2021) | 1319 | 48 |
| | HotpotQA (Yang et al., 2018) | 7,405 | 43 |
| | MATH (Hendrycks et al., 2021) | 5,000 | 100 |
| | MMLU (Hendrycks et al., 2020) | 13,985 | 76 |
| | TheoremQA (Chen et al., 2023a) | 800 | 49 |
| **Total** | | 29,307 | 586 |

across different interaction limits $k$. For each turn, the LLM can choose to interact with tools (generate code to call equation-solver in Fig. 1) or propose a solution (5th turn in Fig. 1). To keep the LLM from getting stuck in an infinite loop of tool-calling without proposing a solution, MINT reminds the LLM: "You have X steps left and Y chances to propose solution left," and provides an additional instruction at the last turn: "You should take the last step to propose a solution." Intuitively, with more interaction with tools, the LLM can get more useful observations through the Python interpreter (e.g., calculation results, error messages). We vary $k \in \{1, 2, 3, 4, 5\}$ and compare the models' success rate with each $k$. We consider LLM's performance gain w.r.t. $k$ and the absolute performance at $k = 5$ as their **tool-augmented task-solving ability** (§3.2).

**Informative user-LLM interaction with language feedback.** Beyond lazy User-LLM interaction, we investigate how the LLM performs when the user mirrors a patient teacher who provides useful suggestions (e.g., technical users who understand the problem-solving process). However, collecting human language feedback for LLM evaluation presents reproducibility challenges due to inconsistent standards and can be costly, particularly for open-source communities with relatively fewer resources[4]. To address these issues, we prompt GPT-4 (§F.4.2) to simulate user language feedback (dotted boxes in Fig. 1). We validate the effectiveness of GPT-4 feedback in a human evaluation (§3.6). We compare the performance between (1) simulated language feedback and (2) lazy user-LLM interaction, both in the setting of tool-augmented interaction with an interaction limit $k = 5$. We consider performance (absolute) and improvements from language feedback as LLM's **ability to leverage natural language feedback**.

## 2.2 REPURPOSING EXISTING DATASETS FOR MINT

Evaluating LLMs in multi-turn interaction can be costly due to the need for iterative inference. For instance, HotpotQA (Yang et al., 2018) has 7,405 test examples. Evaluation with five turns requires at least $7,405 \times 5 = 37K$ LLM inference runs. Previous methods (Yao et al., 2022; Shinn et al., 2023) choose to evaluate on randomly drawn test examples, increasing the barriers to fair comparisons. We select **diverse** tasks from established datasets that **requires multi-turn interaction to solve** while also maintaining the selected subset **compact** for accessible evaluation. The following paragraph describes our three-step approach to repurposing datasets for MINT. We provide dataset sources and statistics in Tab. 1. For more details, please refer to §D in Appendix.

**Data sources of MINT.** Our primary goal is to create a comprehensive evaluation covering tasks that benefit from interaction. We choose three types of tasks:

- Reasoning, including math reasoning (GSM8K, MATH, TheoremQA), multi-hop question answering (HotpotQA), and knowledge problem-solving (MMLU). We implicitly filter out knowledge-intensive questions that do not require multi-step reasoning in the next step.
- Code generation, including HumanEval and MBPP.
- Decision-making tasks in ALFWorld, an embodied household simulator with a text-only interface based on TextWorld (Côté et al., 2018).

---

[4]Based on our human evaluation (§3.6, §B), we estimate annotators, on average, take 96 seconds to provide language feedback for *one turn*, which translates to 90 USD per 100 feedback with hourly wage of US workers.

From eight datasets, we create an initial test set of 29,307 instances. All instances are initially designed for single-round evaluation without interaction, except for decision-making (ALFWorld). Similarly to Yao et al. (2022); Gao et al. (2023), we adapt reasoning tasks into multi-turn interaction tasks by augmented LLM with tools for problem-solving (§F.5.3). Through prompting (§F.5.2), we encourage LLMs to use the Python interpreter to test their generated code on the provided public test suite for code generation problems before committing to a solution.

**Keeping instances that require multi-turn interaction.** To better answer our research question "how LLM benefits from multi-turn interaction," we only keep instances that are challenging and require multi-turn interaction. Since we allow LLM to propose solutions more than once, we filter out instances that a random guess baseline can do well, e.g., multiple-choice instances with $< 4$ options. We then run `gpt-3.5-turbo-0613` (OpenAI API) on the initial dataset and exclude instances finished within two turns (e.g., easy problems that can be solved without multi-turn).

**Stratified sub-sampling for efficient evaluation.** We use stratified sampling (Neyman, 1992) to create a representative set of 586 examples, ensuring that the ratio of correct to incorrect examples in the resulting set mirrors that of the original data to balance the difficulty of the resulting samples.

## 3 EXPERIMENTS

### 3.1 SETUP

**Evaluated LLMs.** To comprehensively measure multi-turn interaction capability and identify the potential gap between open- and closed-source LLMs, we evaluate 4 closed- and 16 open-source LLMs. We cover different sizes and training techniques to better understand how they affect LLMs' multi-turn interaction capability. Training techniques lead to three model variants: pre-trained (base) models, supervised instruction fine-tuned (SIFT, Wei et al., 2022) models, and models trained with reinforcement learning from human feedback (RLHF, Ouyang et al., 2022a). For closed-source models, we evaluate popular commercial LLMs, including `gpt-3.5-turbo-0613` from OpenAI API; `claude-instant-1`, `claude-2` from Anthropic Claude API[5]; Bard `chat-bison-001` from Bard API. For open-source LLMs, we evaluate the `LLaMA-2` model family (7B, 13B, 70B) (Touvron et al., 2023), including `base` and `chat` (RLHF); `Vicuna-v1.5` (7B, 13B) (Zheng et al., 2023), a SIFT model fine-tuned on multi-turn conversations based on `LLaMA-2-base`; the `CodeLLaMA` model family (7B, 13B, 34B) (Rozière et al., 2023) that pre-train `LLaMA-2-base` on code, including `base` and `instruct` (SIFT); `Lemur-v1-70B` (Xu et al., 2023) pre-train `LLaMA-2` on code intensive data, including `base` and `chat` (SIFT).

**Metric.** We consider **Success Rate** $SR$ as our evaluation metric, which measures the percentage of successful task instances. For interaction limit $k$, we start from scratch and allow each LLM to interact up to the $k$-th turn and measure their corresponding $SR_k$. Unless otherwise noted, we limit $k \in [1, 5]$ where $k = 1$ means no interaction and $k = 5$ maximizes interaction turns within most modern LLMs' context window (4,096 tokens).

### 3.2 MEASURING LLM'S TOOL-AUGMENTED TASK-SOLVING IN MULTI-TURN INTERACTION

We ask LLMs to solve tasks (§2.2) with different interaction limits $k \in \{1, 2, 3, 4, 5\}$ *without* natural language feedback (Fig. 1 without red dotted box), and quantify LLMs' tool-augmented task-solving capability by (1) absolute performance $SR_5$ and (2) improvement per additional interaction turn $\Delta_{\text{tools}}$ estimated as the slope $b$ from least-square regression $\min_{b,a} \sum_k (b \cdot k + a - SR_k)^2$ (Tab. 2). Since the underlying $SR_k$ vs. $k$ relationship might not be linear, we only use the regression coefficient (with $R^2$) as a rough estimate of the improvement rate to complement the absolute success rate $SR_5$ for a more comprehensive understanding of the models' capabilities.

**Overall observations.** In Fig. 2, we find all open-source models fall behind best commercial closed-source models in both $SR_5$ and $\Delta_{\text{tools}}$, with `claude-2` and `claude-instant-1` surpassing all open-source LLMs in $\Delta_{\text{tools}}$ with high $R^2$, suggesting near-linear improvement. Notably, despite performing badly at $k = 1$, `claude-instant-1` surpasses `claude-2` as $k$ increases to 3, even-

---

[5]According to https://docs.anthropic.com/claude/reference/selecting-a-model, we use version v1.2 for `claude-instant-1` and v2.0 for `claude-2`.

Table 2: Tool-augmented task-solving success rate with different interaction limit $k$ (i.e., max number of interaction turns allowed) and improvement rate (estimated with least-square regression coefficient, regression $R^2$ is also included). The slope (i.e., coefficient) indicates the rate of improvement while $R^2$ denotes the goodness of fit of the regression model to the data.

| Models | Size | Type | SR (Micro-averaged across tasks) | | | | | Improvement Rate | |
|---|---|---|---|---|---|---|---|---|---|
| | | | $k=1$ | $k=2$ | $k=3$ | $k=4$ | $k=5$ | Slope | $R^2$ |
| **Open-source LLM** | | | | | | | | | |
| CodeLLaMA | 7B | Base* | 0.3 | 4.1 | 7.2 | 7.2 | 4.3 | +1.1 | 0.38 |
| | | SIFT | 0.3 | 7.8 | 10.2 | 9.7 | 8.7 | +1.9 | 0.53 |
| | 13B | Base | 0.5 | 13.7 | 17.9 | 19.3 | 18.4 | +4.1 | 0.70 |
| | | SIFT* | 1.5 | 12.6 | 13.1 | 15.0 | 14.5 | +2.8 | 0.64 |
| | 34B | Base | 0.2 | 16.2 | 23.0 | 25.9 | 28.2 | +6.6 | 0.85 |
| | | SIFT*† | 2.6 | 10.1 | 14.7 | 15.4 | 17.1 | +3.4 | 0.86 |
| LLaMA-2 | 7B | Base | 0.2 | 5.6 | 7.3 | 8.9 | 9.7 | +2.2 | 0.87 |
| | | RLHF* | 1.0 | 4.3 | 6.7 | 6.5 | 7.3 | +1.5 | 0.83 |
| | 13B | Base | 0.2 | 11.4 | 15.5 | 15.2 | 14.5 | +3.2 | 0.63 |
| | | RLHF | 4.1 | 12.5 | 12.5 | 13.3 | 11.9 | +1.7 | 0.47 |
| | 70B | Base | 1.9 | 19.4 | 24.6 | 26.4 | 26.4 | +5.6 | 0.73 |
| | | RLHF | 4.3 | 14.3 | 15.7 | 16.6 | 17.9 | +3.0 | 0.73 |
| Lemur-v1 | 70B | Base | 1.0 | 17.9 | 23.6 | 25.3 | 26.3 | +5.8 | 0.77 |
| | | SIFT | 3.8 | 27.0 | 35.7 | 37.5 | 37.0 | +7.7 | 0.73 |
| Vicuna-v1.5 | 7B | SIFT† | 0.0 | 6.7 | 12.3 | 15.4 | 12.6 | +3.4 | 0.77 |
| | 13B | SIFT† | 0.0 | 2.2 | 4.4 | 6.7 | 8.4 | +2.1 | 1.00 |
| **Closed-source LLM** | | | | | | | | | |
| chat-bison-001 | - | -* | 0.3 | 15.9 | 14.2 | 13.0 | 14.5 | +2.5 | 0.40 |
| claude-2 | - | - | 26.4 | 35.5 | 36.0 | 39.8 | 39.9 | +3.1 | 0.81 |
| claude-instant-1 | - | - | 12.1 | 32.2 | 39.2 | 44.4 | 45.9 | +8.0 | 0.84 |
| gpt-3.5-turbo-0613 | - | - | 2.7 | 16.9 | 24.1 | 31.7 | 36.2 | +8.2 | 0.96 |
| gpt-4-0613 | - | - | - | - | - | - | 69.5 | - | - |

\* Evaluated LLM failed to produce parsable output as instructed in some cases. See §3.5 and Tab. A.7 for details.
† We identified potential undesired artifacts in its training data, which hurt its performance. See §3.5 for details.

tually achieving a higher $SR_5$ (45.9% vs. 39.9%), suggesting `claude-instant-1`'s superior ability to improve with multi-turn interaction.

**Absolute performance and improvement-per-turn scale with model size.** For open-source `CodeLLaMA` and `LLaMA-2`, we observe a trend on all variants (Base, SIFT, and RLHF) that $\Delta_{\text{tools}}$ and $SR_5$ increase when scaling up LLMs. As we discuss in §3.5, `Vicuna-v1.5` models are an exception, potentially due to their training artifacts that hurt task performance.

**SIFT on multi-turn data can be helpful.** Despite the issue above, `Vicuna-v1.5` (7B, SIFT) does show stronger performance compared to `LLaMA-2` (Base and RLHF, 7B) in $\Delta_{\text{tools}}$ (+3.4% vs. +2.2% / +1.5%) and $SR_5$ (12.6% vs. 9.7% / 7.3%). `Lemur-v1` (70B, SIFT) also shows stronger performance than its Base variant. However, except `CodeLLaMA` (7B), we do not find similar improvements on `CodeLLaMA` (SIFT). We hypothesize that the performance gain on `Vicuna-v1.5` and `Lemur-v1` could be attributed to fine-tuning on ShareGPT's multi-turn human-ChatGPT conversations.

**RLHF could hurt LLM-tool multi-turn interaction.** We find that on `LLaMA-2` series, RLHF alignment generally hurts models' performance in both $\Delta_{\text{tools}}$ (−0.7% to −2.6%) and $SR_5$ (−2.4% to −8.5%), similar to the

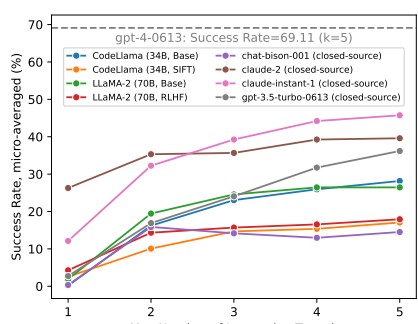

Figure 2: With an increasing interaction limit $k$, the success rate (%, micro averaged) improves at different rates for different LLMs. For clarity, only a subset of evaluated LLMs are visualized.

prior observation that alignment can degrade task performance (Ouyang et al., 2022b). However, it's hard to conclude that RLHF in general hurts model performance. We leave it for future work to explore the role of RLHF in multi-turn interaction.

### 3.3 Measuring LLM's Ability to Leverage Natural Language Feedback

On top of LLM-tool interaction, we use `gpt-4-0613` to simulate user feedback for evaluated LLMs (Fig. 1 with red dotted box). With a $k = 5$ interaction limit, we measure the LLM's ability to leverage natural language feedback using the absolute performance $SR_5^{\text{feedback}}$ and the performance difference after feedback is given: $\Delta_{\text{feedback}} = SR_5^{\text{feedback}} - SR_5$. We present results in Tab. 3.

**Overall observations.** We find no significant difference between open- and closed-source models in terms of $\Delta_{\text{feedback}}$. Open-source models obtain $+1.7 - +17.2\%$ from feedback, while closed-source models obtain $+6.5 - +15.2\%$. However, there is still a gap between them in absolute success rate $SR_5^{\text{feedback}}$, as the best open-source model `Lemur-v1` (70B, SIFT) still lags behind the best closed-source model `claude-instant-1` by $8.7\%$. Surprisingly, we find that `CodeLLaMA-34B-base` can achieve comparable performance to GPT-4 on decision-making tasks with language feedback from it, showing its strong ability to leverage language feedback.

**The effect of SIFT and RLHF.** Similar to §3.2, we find that SIFT and RLHF hurt models' ability to leverage feedback. The results on `CodeLLaMA` (except 7B) and `LLaMA-2` show that SIFT/RLHF models all have lower $\Delta_{\text{feedback}}$ and $SR_5^{\text{feedback}}$ than their base variants. Another two exceptions are `Vicuna-v1.5` (7B) and `Lemur-v1` (70B). We speculate using multi-turn conversations (ShareGPT) for SIFT contributes to these two exceptions.

### 3.4 Measuring the Efficacy of Different LLM's Ability to Provide Feedback

Fixing the evaluated model to be `gpt-3.5-turbo-0613`, we assess seven LLMs' feedback-providing capability through $\Delta_{\text{feedback}}$ (Tab. 4). Our main finding is that task-solving ability could be orthogonal to feedback-providing ability: LLM's higher task-solving performance does not guarantee better feedback-providing capability and vice versa. For example, although GPT-3.5 (16k) performs well in task-solving ($SR_5$ ranked 3rd in Tab. 4), it leads to a performance degradation of $-10.4\%$ in GPT-3.5; Similarly, GPT-4 with self-feedback in Tab. 3 also experiences degraded performance. On the other hand, despite performing the worst in solving tasks in Tab. 4, `CodeLLaMA-34B-Instruct` can provide feedback that improves the stronger GPT-3.5.

### 3.5 MINT Can Help Detect Failure Patterns of Evaluated LLMs

Surprisingly, beyond evaluating LLMs' multi-turn interaction ability, we find complex multi-turn tasks in MINT can also test an LLM for unexpected behavior. We find two main types of anomalies: (1) inability to follow formatting instructions and (2) unexpected outputs likely due to artifacts.

**Inability to follow formatting instructions.** We find that some models (e.g., smaller `CodeLLaMA` and `LLaMA`, `chat-bison-001`) have trouble producing a parsable format as instructed, hindering task-solving (statistics can be found in Tab. A.7).

**Unexpected output likely due to data artifact.** We find that Vicuna models (SIFT on ShareGPT data) generate escaped underscore ("\_") instead of underscore ("_") across all tasks, causing syntax errors when executing code and reducing performance. We examine ShareGPT data (2023) and find at least one escaped underscore ("\_") artifact on $15\%$ examples, suggesting artifacts in training data could cause this issue. We observe a similar issue with `CodeLLaMA-Instruct`: We find that `CodeLLaMA-Instruct` (34B) *always* ignores user-given instructions on the code generation tasks "wrap your code with `<execute>` tag" and uses `[PYTHON]` to wrap the code (happens on $100\%$ of code generation tasks, $0\%$ on other tasks). Touvron et al. (2023) uses `[PYTHON]` as the tag to generate self-instruct data on code problems for SIFT. We suspect `CodeLLaMA-Instruct` models are trained and overfitted to `[PYTHON]` token, causing them to produce `[PYTHON]` regardless of user instruction. We refer to §E.1 and §E.2 for examples and quantitative results.

### 3.6 Can GPT-4 Generate Human-Level Natural Language Feedback?

We perform a human evaluation quantitatively comparing the feedback generated by GPT-4 and written by humans. Details can be found in Appendix §B. In Tab. 5, human annotators consider $91.2\%$ of GPT-4 generated language feedback to be as helpful as, if not better than, human written feedback. It's also hard for humans to distinguish GPT-4 generated feedback from human feedback

Table 3: LLM's ability to leverage natural language feedback, measured by $\Delta_{\texttt{feedback}}$ between models' performance with and without feedback produced by `gpt-4-0613`. All models are evaluated with an interaction turn limit of $k = 5$. For both open- and closed-source LLMs, the best performance is bolded, and the second-best performance is underlined.

| Evaluated LLM | Size | Type | Setup | Reasoning | Decision-Making | Code | Micro Average |
|---|---|---|---|---|---|---|---|
| | | | **Open-source LLM** | | | | |
| CodeLLaMA | 7B | Base | no feedback | *0.0 | 18.7 | *0.0 | 4.3 |
| | | | w/ GPT-4 feedback | 4.8 | 59.7 | 0.0 | 16.2 |
| | | | $\Delta_{\texttt{feedback, gpt-4}}$ | +4.8 | +41.0 | +0.0 | +11.9 |
| | | SIFT | no feedback | 7.9 | 17.2 | 2.2 | 8.7 |
| | | | w/ GPT-4 feedback | 17.1 | 62.7 | 10.3 | 25.9 |
| | | | $\Delta_{\texttt{feedback, gpt-4}}$ | +9.2 | +45.5 | +8.1 | +17.2 |
| | 13B | Base | no feedback | 8.5 | 56.0 | 4.4 | 18.4 |
| | | | w/ GPT-4 feedback | 15.8 | 73.9 | 27.9 | 31.9 |
| | | | $\Delta_{\texttt{feedback, gpt-4}}$ | +7.3 | +17.9 | +23.5 | +13.5 |
| | | SIFT | no feedback | 4.8 | 50.0 | †2.2 | 14.5 |
| | | | w/ GPT-4 feedback | 10.1 | 59.0 | 14.7 | 22.4 |
| | | | $\Delta_{\texttt{feedback, gpt-4}}$ | +5.4 | +9.0 | +12.5 | +7.8 |
| | 34B | Base | no feedback | 17.4 | **63.4** | 18.4 | 28.2 |
| | | | w/ GPT-4 feedback | 30.4 | **84.3** | 30.1 | 42.7 |
| | | | $\Delta_{\texttt{feedback, gpt-4}}$ | +13.0 | +20.9 | +11.8 | +14.5 |
| | | SIFT | no feedback | 14.9 | 37.3 | *†2.2 | 17.1 |
| | | | w/ GPT-4 feedback | 20.2 | 67.9 | 3.7 | 27.3 |
| | | | $\Delta_{\texttt{feedback, gpt-4}}$ | +5.4 | +30.6 | +1.5 | +10.2 |
| LLaMA-2 | 7B | Base | no feedback | 2.9 | 35.8 | 0.0 | 9.7 |
| | | | w/ GPT-4 feedback | 4.1 | 46.3 | 8.1 | 14.7 |
| | | | $\Delta_{\texttt{feedback, gpt-4}}$ | +1.3 | +10.5 | +8.1 | +4.9 |
| | | RLHF | no feedback | 13.6 | *0.0 | 0.0 | 7.3 |
| | | | w/ GPT-4 feedback | 14.6 | 2.2 | 2.9 | 9.0 |
| | | | $\Delta_{\texttt{feedback, gpt-4}}$ | +1.0 | +2.2 | +2.9 | +1.7 |
| | 13B | Base | no feedback | 3.5 | 50.0 | 5.2 | 14.5 |
| | | | w/ GPT-4 feedback | 10.8 | 60.5 | 15.4 | 23.2 |
| | | | $\Delta_{\texttt{feedback, gpt-4}}$ | +7.3 | +10.5 | +10.3 | +8.7 |
| | | RLHF | no feedback | 19.6 | 3.7 | 2.2 | 11.9 |
| | | | w/ GPT-4 feedback | 24.1 | 9.7 | 10.3 | 17.6 |
| | | | $\Delta_{\texttt{feedback, gpt-4}}$ | +4.4 | +6.0 | +8.1 | +5.6 |
| | 70B | Base | no feedback | 18.7 | 59.0 | 12.5 | 26.4 |
| | | | w/ GPT-4 feedback | 22.5 | 73.1 | 27.9 | 35.3 |
| | | | $\Delta_{\texttt{feedback, gpt-4}}$ | +3.8 | +14.2 | +15.4 | +8.9 |
| | | RLHF | no feedback | 20.2 | 21.6 | 8.8 | 17.9 |
| | | | w/ GPT-4 feedback | 23.1 | 41.8 | 19.9 | 26.6 |
| | | | $\Delta_{\texttt{feedback, gpt-4}}$ | +2.9 | +20.1 | +11.0 | +8.7 |
| Lemur-v1 | 70B | Base | no feedback | 16.1 | 61.2 | 15.4 | 26.3 |
| | | | w/ GPT-4 feedback | 20.9 | 70.2 | 27.9 | 33.8 |
| | | | $\Delta_{\texttt{feedback, gpt-4}}$ | +4.8 | +9.0 | +12.5 | +7.5 |
| | | SIFT | no feedback | **31.6** | 59.7 | **27.2** | **37.0** |
| | | | w/ GPT-4 feedback | **32.6** | 68.7 | **44.9** | **43.7** |
| | | | $\Delta_{\texttt{feedback, gpt-4}}$ | +0.9 | +9.0 | +17.6 | +6.7 |
| Vicuna-v1.5 | 7B | SIFT | no feedback | †10.1 | 29.1 | †2.2 | 12.6 |
| | | | w/ GPT-4 feedback | 9.8 | 64.9 | 6.6 | 21.7 |
| | | | $\Delta_{\texttt{feedback, gpt-4}}$ | −0.3 | +35.8 | +4.4 | +9.0 |
| | 13B | SIFT | no feedback | †11.1 | †8.2 | †2.2 | 8.4 |
| | | | w/ GPT-4 feedback | 16.5 | 5.2 | 1.5 | 10.4 |
| | | | $\Delta_{\texttt{feedback, gpt-4}}$ | +5.4 | −3.0 | −0.7 | +2.1 |
| | | | **Closed-source LLM** | | | | |
| chat-bison-001 | - | - | no feedback | *14.2 | 29.9 | *0.0 | 14.5 |
| | | | w/ GPT-4 feedback | 25.0 | 47.0 | 6.6 | 25.8 |
| | | | $\Delta_{\texttt{feedback, gpt-4}}$ | +10.8 | +17.2 | +6.6 | +11.3 |
| claude-2 | - | - | no feedback | **52.2** | 14.2 | **36.8** | 39.9 |
| | | | w/ GPT-4 feedback | **55.1** | 41.0 | **47.1** | 50.0 |
| | | | $\Delta_{\texttt{feedback, gpt-4}}$ | +2.8 | +26.9 | +10.3 | +10.1 |
| claude-instant-1 | - | - | no feedback | 50.0 | 47.0 | 35.3 | **45.9** |
| | | | w/ GPT-4 feedback | 54.4 | 53.0 | 47.1 | **52.4** |
| | | | $\Delta_{\texttt{feedback, gpt-4}}$ | +4.4 | +6.0 | +11.8 | +6.5 |
| gpt-3.5-turbo-0613 | - | - | no feedback | 36.7 | 41.8 | 29.4 | 36.2 |
| | | | w/ GPT-4 feedback | 50.3 | **66.4** | 39.0 | 51.4 |
| | | | $\Delta_{\texttt{feedback, gpt-4}}$ | +13.6 | +24.6 | +9.6 | +15.2 |
| gpt-4-0613 | - | - | no feedback | 67.4 | 84.3 | 59.6 | 69.5 |
| | | | w/ GPT-4 feedback | 67.1 | 85.1 | 56.6 | 68.8 |
| | | | $\Delta_{\texttt{feedback, gpt-4}}$ | −0.3 | +0.7 | −2.9 | −0.7 |

\* Evaluated LLM failed to produce parsable output as instructed in some cases (§2.1). See §3.5 and Tab. A.7 for details.
† We identified potential undesired artifacts in its training data, which hurt its performance. See §3.5 for details.

Table 4: LLMs' ability to provide feedback, measured by $\Delta_{\text{feedback}}$ with a fixed evaluated LLM (GPT-3.5). We also report $SR_5$ differences between the feedback-provider and evaluated LLM.

| Feedback-provider LLM | $SR_5$ Difference | $\Delta_{\text{feedback}}$ |
|---|---|---|
| `gpt-4-0613` | +33.3 | +15.2 |
| `claude-instant-1` | +9.7 | +1.5 |
| `gpt-3.5-turbo-16k-0613` | +4.1 | −10.4 |
| `CodeLlama-34b` (Base) | −8.0 | +2.4 |
| `Llama-2-70b` (Base) | −9.7 | −0.5 |
| `Llama-2-70b-chat` (RLHF) | −18.3 | −14.0 |
| `CodeLlama-34b-Instruct` (SIFT) | −19.1 | +3.2 |

Table 5: Human Evaluation of GPT-4 Generated Feedback against human written feedback, measuring helpfulness and human-like.

| | Percentage (%) | |
|---|---|---|
| **Which feedback is more** | Helpful | Human-Like |
| Both are equally | 36.3 | **69.9** |
| GPT-4 feedback | **54.9** | 22.1 |
| Human feedback | 8.8 | 8.0 |

(human-like) in 92% of the cases. We also compare GPT-4 generated and human-written feedback by asking `gpt-3.5-turbo-0613` to continue problem-solving with either a turn of (1) human language feedback or (2) GPT-4 feedback. Results show that human feedback and GPT-4 feedback lead to similar model performance $SR_5^{\text{feedback}}$ (32.7% vs. 33.6%).

## 4 RELATED WORK

### 4.1 LLM IN INTERACTION

**Interact with users.** LLMs have demonstrated extensive potential in seamless interaction with human users and in assimilating real-time human feedback during inference processes (Fernandes et al., 2023). According to recent studies, this collaborative synergy between humans and LLMs has been explored across various domains and applications, including sentence editing (Reid & Neubig, 2022; Schick et al., 2023c; Chen et al., 2023b), semantic parsing (Yan et al., 2023; Yao et al., 2019), code generation (Nijkamp et al., 2023; Elgohary et al., 2020), iterative output refinement (Saunders et al., 2022), and creative writing (Lee et al., 2022a; Shu et al., 2023; Wang et al., 2023b), generative information-seeking (Kamalloo et al., 2023), and even theorem proving (Yang et al., 2023b). The partnership between users and LLMs continues to redefine possibilities across diverse research areas.

**Interact with tools.** Engaging with external tools allows LLMs can lead to more accurate and reliable outputs (Peng et al., 2023; Gou et al., 2023; Qin et al., 2023a). LLMs can be connected with real-world Application Programming Interfaces (APIs), enabling them to actively engage with diverse external tools (Qin et al., 2023b; Parisi et al., 2022; Schick et al., 2023a; Tang et al., 2023; Patil et al., 2023; Song et al., 2023; Hao et al., 2023; Yuan et al., 2023). For example, LLMs can connect with (1) the Internet to obtain latest information (Nakano et al., 2021; Shuster et al., 2022; Paranjape et al., 2023; Liu et al., 2023b); (2) the program interpreter to run the generated code (Chen et al., 2022; Gao et al., 2023; Drori et al., 2022; Pan et al., 2023; Wang et al., 2023a; 2024); (3) multimodal perceiver to obtain the information beyond the language modality (Huang et al., 2023a; Lu et al., 2023); (4) physical simulator to better understand the physical law (Liu et al., 2023a).

### 4.2 EVALUATING INTERACTION

Existing work on interaction evaluation mostly focuses on a specific task or dimension, like task completion (Liu et al., 2023c), code generation (Yang et al., 2023a), human-LLM collaborative task solving (Lee et al., 2022b; Huang et al., 2023b; Fu et al., 2023), tool manipulation (Tang et al., 2023), and web nevigation (Zhou et al., 2023; Deng et al., 2023a). That is, they solely focus on interacting with either the environment or humans, often on a specific task, overlooking the fundamental importance of both elements in LLM interaction. Different from prior work, MINT covers a range of diverse tasks and is designed to measure the multi-turn interaction capabilities of LLMs with both tools and user feedback that are more aligned with real-world applications.

## 5 CONCLUSION

In this work, we present MINT, an evaluation benchmark designed to evaluate LLM's task-solving ability in multi-turn interaction by using tools and leveraging natural language feedback, which we simulate using GPT-4. We hope MINT can serve as a helpful resource to help track progress and incentivize future research in improving LLM's multi-turn task-solving capabilities. We refer to §A for a discussion of limitations and future work.

## ACKNOWLEDGEMENT

We thank the anonymous reviewers for their suggestions and comments. This research is based upon work supported by U.S. DARPA ECOLE Program No. HR00112390060 and ITM Program No. FA8650-23-C-7316 and KAIROS Program No. FA8750-19-2-1004. The views and conclusions contained herein are those of the authors and should not be interpreted as necessarily representing the official policies, either expressed or implied, of DARPA, or the U.S. Government. The U.S. Government is authorized to reproduce and distribute reprints for governmental purposes notwithstanding any copyright annotation therein. HP is in part supported by a gift from Apple.

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

## A  LIMITATIONS AND FUTURE WORK

We simulate the natural language feedback of human users with GPT-4. Despite showing in a human experiment that it is similar to human-written feedback, however, GPT-4 simulated might not cover all the possible responses from real-human users and may not suitably simulate every aspect of human feedback, particularly in tasks (e.g., policy-making) that involve nuanced judgments of human values. While the focus of our work lies on LLM's in-context multi-turn interaction, we have yet to explore the potential of directly leveraging language feedback for model training and improvement similar to Wang et al. (2023a), which we leave for future work. Furthermore, our metrics may not fully assess the quality of the interaction process beyond outcomes. For example, models repetitively guessing to get higher scores should be penalized. Despite our best efforts to ensure our benchmark contains challenging and comprehensive tasks, there is still a wide range of tools (Qin et al., 2023c) and real-world use cases (e.g., web-browsing Deng et al. (2023b), operating system Liu et al. (2023d)) that MINT did not cover. Instead of making this benchmark a one-time effort, we hope to continuously improve this benchmark by integrating more challenging tasks and tools as LLMs get better.

## B  DETAILS OF HUMAN EVALUATION

We perform two stages of human annotation using the Potato annotation interface (Pei et al., 2022). In the first stage, we ask two human annotators (A and B) to provide language feedback for a trajectory. We randomly sample 2 instances of interaction trajectories per task from a subset of 8 evaluated LLMs to maximize diversity (in Tab. 3). We filter out task instances that succeed in the first turn (i.e., no need for feedback), resulting in 113 interaction trajectories for annotation. We randomly select a turn for each task trajectory and remove all interactions and GPT-4 generated feedback after that turn. We randomly divide the 113 instances into two subsets and assign each subset to one human annotator. Given previous interaction history, human annotators A and B are asked to provide a turn of natural language feedback as if interacting with ChatGPT. Annotation of each feedback, on average, takes 96 seconds. According to US Bureau of Labor Statistics (2023), U.S. private non-farm worker average about $33.82 hourly wage (Aug 2023), which translate to an annotation cost of $90 per 100 turns of feedback.

In the second stage, we ask two different human annotators (C and D) to compare human-annotated feedback (from the first stage) and GPT-4 generated feedback (from the original trajectory) on two dimensions: helpfulness and human-like. Specifically, helpfulness means whether feedback is helpful for the LLM to succeed in this task, while human-like focuses on the literal similarity of feedback and human usage. For each dimension, we ask them to determine which feedback is better (i.e., more helpful or human-like) or both are equally good.

## C  ABLATION STUDY

### C.1  HOW DO FEEDBACK VARIATIONS IMPACT FEEDBACK QUALITY $\Delta_{\text{FEEDBACK}}$?

To gain deeper insights into the effects of various feedback settings on enhancing the performance of language models, we perform an ablation study on feedback by controlling feedback *informativeness* and *frequency*. See §F.4.2 for detailed implementation. We present the results in Tab. A.6.

#### C.1.1  INFORMATIVENESS

We define informativeness in two dimensions: (1) whether the generated feedback is conditioned on the ground-truth solution (**w/ GT**) or not (**w/o GT**, default setting); (2) whether the feedback provided to LLM is **textual** (default setting) or **binary** (i.e., good vs. bad).

**Conditioned on Ground-truth Information**  In Tab. C.1.1, we find that adding ground-truth information into the feedback generator improves the quality of feedback for reasoning and code generation. However, this trend doesn't hold for decision-making, where using ground-truth information for feedback leads to a performance drop ($-8.95\%$) compared to no feedback. We hypothesize that this discrepancy can be attributed to the unique nature of decision-making tasks. Unlike other tasks

Table A.6: Ablation of different factors (informativeness, frequency) that impact feedback quality, using `gpt-3.5-turbo-0613` as the evaluated LLM and `gpt-4-0613` to simulate language feedback.

| Setup | Reasoning | Decision Making | Code Generation | Micro Average |
|---|---|---|---|---|
| w/o feedback | 36.25 | 41.79 | 29.41 | 35.93 |
| $\Delta_{\text{feedback, textual, w/o GT, dense}}$ | +13.44 | +24.63 | +9.56 | +15.09 |
| *Informativeness of Feedback* | | | | |
| $\Delta_{\text{feedback, w/ GT}}$ | +16.87 | −8.95 | +18.38 | +11.36 |
| $\Delta_{\text{+GT feedback}}$ | +3.43 | −33.58 | +8.82 | −3.73 |
| $\Delta_{\text{feedback, binary}}$ | +2.19 | +5.97 | +0.74 | +2.71 |
| $\Delta_{\text{−textual feedback}}$ | −11.25 | −18.66 | −8.82 | −12.38 |
| *Frequency of Feedback* | | | | |
| $\Delta_{\text{feedback, sparse}}$ | +5.31 | +4.48 | +0.74 | +4.07 |
| $\Delta_{\text{−feedback frequency}}$ | −8.13 | −20.15 | −8.82 | −11.02 |

with definitive solutions, decision-making tasks involve generating action trajectories as solutions (e.g., §F.6). When the initial actions of the model deviate from the ground-truth trajectory, comparing its actions with the ground-truth actions could confuse the feedback-provider LLM, resulting in suboptimal feedback quality.

**Provide Binary Feedback**  We find that providing LLM with binary feedback (i.e., a binary label of good or bad) instead of more informative textual feedback (i.e., a superset of binary feedback) inevitably hurts performance on all tasks. However, we observe that binary feedback alone provides performance benefits compared to no feedback, especially for decision-making (+5.97), where early action can profoundly impact final task success. In these cases, providing step-wise binary feedback can help LLM agents terminate bad initial actions and backtrack, leading to a higher task success rate.

### C.1.2 FREQUENCY

We investigate the role of feedback frequency: whether we are providing feedback to the LLM every step (**Dense**) or only when the LLM agent proposes a solution (**Sparse**, i.e., when the LLM thinks it finishes the task).

In Tab. A.6, as expected, we find changing from dense to sparse feedback hurts performance (−11.02 on average). However, we observe positive performance gain on all tasks, similar to binary feedback (§C.1.1), suggesting that sparse feedback alone is valuable. Note that when evaluating on sparse feedback setting, MINT is equivalent to the setting of Reflexion feedback (Shinn et al., 2023).

## D  DATASET FILTERING AND DOWN-SAMPLING

The dataset curation can be summarized into three steps:

**Collect data from the test set of 8 different datasets shown in Table 1.**

For HotpotQA we reserve the first 500 instances. Then, we format dataset prompts with (`'Task:'`, `task_description`, `solution_range`). For the solution_range variable, in GSM8K it is set to be integer, and in TheoremQA it is set corresponding to the instance requirement (float, integer, list of integers, option). For other datasets, since they don't have a specific solution range requirement, we set solution_range to be an empty string. An example from TheoremQA is as follows:

```
Task:  Let M be the inverse of the group element ((3, 5), (4, 6))
in Z_7.  What is M[0][1]?  Output format required:  integer.
```

In this example, task_description is: "Let M be the inverse of the group element ((3, 5), (4, 6)) in Z_7. What is M[0][1]?" and solution_range is: "Output format required: integer."

Table A.7: The average number of interaction turns an LLM failed due to not following the instructed format (i.e., not producing `<execute>` or `<solution>` tag as instructed under $k = 5$ and no feedback setting, §2.1) vs. the average number of total turns. All LLMs that produce more than 20% of such invalid actions w.r.t total turns are bolded.

| Evaluated LLM | Size | Type | Number of Failed Turn due to Format: Total Turns (Averaged) | | | |
| | | | Reasoning | Decision | Code | Micro Average |
|---|---|---|---|---|---|---|
| | | | **Open-source LLM** | | | |
| CodeLlama | 7B | Base | **3.96 / 4.99** | 0.11 / 4.17 | **2.38 / 4.38** | **2.71 / 4.66** |
| | | SIFT | 0.46 / 4.32 | 0.10 / 4.33 | 0.10 / 4.65 | 0.30 / 4.40 |
| | 13B | Base | 0.50 / 4.55 | 0.00 / 3.36 | 0.00 / 4.93 | 0.27 / 4.36 |
| | | SIFT | 0.16 / 4.66 | 0.01 / 3.77 | 0.04 / 4.77 | 0.10 / 4.48 |
| | 34B | Base | 0.19 / 4.21 | 0.00 / 3.37 | 0.05 / 4.77 | 0.11 / 4.15 |
| | | SIFT | 0.23 / 3.68 | 0.04 / 3.83 | **1.09 / 3.27** | 0.39 / 3.62 |
| LLaMA-2 | 7B | Base | 0.59 / 4.62 | 0.00 / 3.53 | 0.25 / 4.96 | 0.38 / 4.45 |
| | | RLHF | 0.75 / 4.03 | **1.13 / 4.40** | 0.72 / 3.79 | **0.83 / 4.06** |
| | 13B | Base | 0.49 / 4.75 | 0.01 / 3.40 | 0.13 / 4.96 | 0.30 / 4.49 |
| | | RLHF | 0.29 / 3.71 | 0.00 / 4.54 | 0.10 / 3.02 | 0.18 / 3.74 |
| | 70B | Base | 0.19 / 4.19 | 0.00 / 3.31 | 0.16 / 4.49 | 0.14 / 4.06 |
| Lemur-v1 | 70B | Base | 0.29 / 4.25 | 0.00 / 3.28 | 0.26 / 4.33 | 0.22 / 4.05 |
| | | SIFT | 0.35 / 3.88 | 0.01 / 3.34 | 0.03 / 4.07 | 0.20 / 3.80 |
| Vicuna-v1.5 | 7B | SIFT | 0.26 / 4.64 | 0.06 / 3.54 | 0.02 / 4.78 | 0.16 / 4.42 |
| | 13B | SIFT | 0.08 / 4.80 | 0.49 / 4.66 | 0.07 / 4.90 | 0.17 / 4.79 |
| | | | **Closed-source LLM** | | | |
| chat-bison-001 | - | - | **2.27 / 3.84** | 0.10 / 4.18 | **4.62 / 4.87** | **2.32 / 4.16** |
| claude-2 | - | - | 0.02 / 1.86 | 0.01 / 3.51 | 0.00 / 2.24 | 0.02 / 2.32 |
| claude-instant-1 | - | - | 0.06 / 2.81 | 0.00 / 3.91 | 0.02 / 3.76 | 0.04 / 3.28 |
| gpt-3.5-turbo-0613 | - | - | 0.50 / 4.18 | 0.00 / 3.87 | 0.07 / 4.26 | 0.29 / 4.13 |
| gpt-4-0613 | - | - | 0.04 / 3.11 | 0.00 / 2.87 | 0.00 / 3.42 | 0.02 / 3.13 |

**Keeping instances that requires multi-turn interaction.**

- We first clean up multiple-choice tasks with less than 4 options. These tasks are primarily from MMLU and TheoremQA datasets.
- For MMLU and MATH, since their test sets are large and have various classes of tasks (e.g., for MATH they have algebra, geometry, pre-calculus), we firstly roughly clean those classes that do not need interaction (e.g. for MMLU they have "philosophy" domain which does not need much interaction but only requires some basic knowledge about philosophy) by picking up $N$ instances from each class, run these instances with `gpt-3.5-turbo-0613`, and exclude those classes whose average interaction turn across instances are less than $k$ turns. For math we set $N = 100$ and $k = 3.5$, for MMLU we set $N = 20$ and $k = 2.5$. Remaining classes of MATH: Intermediate Algebra, Pre-calculus, Algebra, Geometry, Number Theory. Remaining classes of MMLU: world religions test, virology test, college mathematics test, astronomy test, college physics test, high school chemistry test, global facts test, high school mathematics test, formal logic test.
- we run all remaining data with `gpt-3.5-turbo-0613` with turn budget $k = 5$, no feedback, and exclude those instances with k≤2.

**Stratified sub-sampling for efficient evaluation.**

After cleaning data, we want to maintain data difficulty and balance different types of tasks while continuing sub-sampling. We stratify the instances based on the dataset and whether `gpt-3.5-turbo-0613` has completed it (i.e., $8 \times 2 = 16$ strata). For each stratum we set different proportions of instances to be preserved: $p_{\text{alfworld}} = 1$, $p_{\text{mbpp}} = p_{\text{humaneval}} = 0.5$, $p_{\text{gsm8k}} = p_{\text{hotpotqa}} = p_{\text{theoremqa}} = 0.2$, $p_{\text{MMLU}} = 0.1$, $p_{\text{MATH}} = 0.05$.

Table A.8: Summary of Tools by Task Type

| Task Type | Tool Signature |
|---|---|
| Decision-Making | ```[1] put(object:  str, receptacle:  str) -> str``` 
 ```[2] goto(receptacle:  str) -> str``` 
 ```[3] take_from(object:  str, receptacle:  str) -> str``` 
 ```[4] open_receptacle(receptacle:  str) -> str``` 
 ```[5] toggle(object_or_receptacle:  str) -> str``` 
 ```[6] close_receptacle(receptacle:  str) -> str``` 
 ```[7] clean(object:  str, receptacle:  str) -> str``` 
 ```[8] heat(object:  str, receptacle:  str) -> str``` 
 ```[9] cool(object:  str, receptacle:  str) -> str``` 
 ```[10] use(receptacle:  str) -> str``` 
 ```[11] look() -> str``` |
| Reasoning | ```[1] wikipedia_search(query:  str) -> str``` |
| Code Generation | ```No tool is provided``` |

# E  ISSUES

## E.1  VICUNA-v1.5 ESCAPE UNDERSCORE ISSUE

The following is a random trajectory (in-context example omitted) from `Vicuna-13b-v1.5` model[6]. For some unknown reason, it tends to escape any underscore ("\_") that appears in the code, causing it to fail some task instances. Quantitatively, we calculate the percentage of turns that contain an escaped underscore over different LLMs in Tab. A.9, and find that this is a particular issue with `Vicuna` and SIFT version of `Lemur-v1`. We checked $94,145$ instances of ShareGPT data[7] and found that about $15\%$ ($14,110$) of the conversations have the pattern of backslash underscore ("\_"). We believe these artifacts in the instruction tuning dataset could be the reason that causes `Vicuna` and `Lemur-v1` to generate code with these backslash underscore patterns ("\_").

```
=== user ===
You are a helpful assistant assigned with the task of problem-solving. To
↪  achieve this, you will be using an interactive coding environment
↪  equipped with a variety of tool functions to assist you throughout
↪  the process.

At each turn, you should first provide your step-by-step thinking for
↪  solving the task. Your thought process should be enclosed using
↪  "<thought>" tag, for example: <thought> I need to print "Hello
↪  World!" </thought>.

After that, you have two options:

1) Interact with a Python programming environment and receive the
↪  corresponding output. Your code should be enclosed using "<execute>"
↪  tag, for example: <execute> print("Hello World!") </execute>.
2) Directly provide a solution that adheres to the required format for
↪  the given task. Your solution should be enclosed using "<solution>"
↪  tag, for example: The answer is <solution> A </solution>.

You have 5 chances to interact with the environment or propose a
↪  solution. You can only propose a solution 2 times.

---
```

[6] https://huggingface.co/lmsys/vicuna-13b-v1.5 
[7] https://huggingface.co/datasets/anon8231489123/ShareGPT_Vicuna_unfiltered

Table A.9: Percentage of task instances that assistant outputs escaped underscore ("\_") under the setting of $k = 5$ without language feedback, which causes Syntax Error during task-solving. This seems to be a particular issue with `Vicuna` LLMs. Percentages greater than 2% are bolded.

| Percentage of assistant interactions that contains escaped underscore (%) | | | | | |
|---|---|---|---|---|---|
| Evaluated LLM | Size | Type | Reasoning | Decision-Making | Code |
| **Open-source LLM** | | | | | |
| `CodeLLaMA` | 7B | Base | 0.00 | 0.00 | 0.00 |
| | | SIFT | 0.00 | 0.00 | 0.00 |
| | 13B | Base | 0.00 | 0.00 | 0.00 |
| | | SIFT | 0.00 | 0.00 | 0.00 |
| | 34B | Base | 0.00 | 0.00 | 0.00 |
| | | SIFT | 0.00 | 0.00 | 0.00 |
| `LLaMA-2` | 7B | Base | 0.00 | 0.00 | 0.00 |
| | | RLHF | 0.00 | 0.00 | 0.00 |
| | 13B | Base | 0.00 | 0.00 | 0.00 |
| | | RLHF | 0.00 | 0.00 | 0.00 |
| | 70B | Base | 0.00 | 0.00 | 0.00 |
| `Lemur-v1` | 70B | Base | 0.00 | 0.00 | 0.00 |
| | | SIFT | **2.22** | 0.00 | 0.00 |
| `Vicuna-v1.5` | 7B | SIFT | **30.38** | 0.75 | **11.03** |
| | 13B | SIFT | **38.61** | **86.57** | **90.44** |
| **Closed-source LLM** | | | | | |
| `chat-bison-001` | - | - | 0.00 | 0.00 | 0.00 |
| `claude-2` | - | - | 0.32 | 0.00 | 0.00 |
| `claude-instant-1` | - | - | 0.00 | 0.00 | 0.00 |
| `gpt-3.5-turbo-0613` | - | - | 0.00 | 0.00 | 0.00 |
| `gpt-3.5-turbo-16k-0613` | - | - | 0.00 | 0.00 | 0.00 |
| `gpt-4-0613` | - | - | 0.00 | 0.00 | 0.00 |

```
Task:
Write a function to check if a nested list is a subset of another nested
↪  list.
assert check_subset_list([1, 2, 3, 4, 5, 6, 7, 8, 9, 10, 11, 12, 13,
↪  14],[[12, 18, 23, 25, 45], [7, 11, 19, 24, 28], [1, 5, 8, 18, 15,
↪  16]])==False

=== assistant ===
Assistant:
<thought> To check if a nested list is a subset of another nested list, I
↪  should iterate through the elements of the first list and check if
↪  they are present in the second list. If all elements are present,
↪  then the first list is a subset of the second list. </thought>
<execute>
def check\_subset\_list(list1, list2):
    for element in list1:
        if element not in list2:
            return False
    return True
res = check\_subset\_list(
    [1, 2, 3, 4, 5, 6, 7, 8, 9, 10, 11, 12, 13, 14],
    [[12, 18, 23, 25, 45],
    [7, 11, 19, 24, 28],
```

```
        [1, 5, 8, 18, 15, 16]]
)
assert res == False, "Expected False but got {}".format(res)
</execute>
```

```
=== user ===
Observation:
Cell In[1], line 1
    def check\_subset\_list(list1, list2):
                                            ^
SyntaxError: unexpected character after line continuation character
You have 4 steps left and 2 chances to propose solution left.
```

```
Expert feedback:
This is BAD. You have a syntax error in your function definition. The
↪   backslashes are not needed in the function name. Also, your logic is
↪   incorrect. You are checking if the elements of the first list are in
↪   the second list, but the second list is a list of lists. You need to
↪   check if the elements of the first list are in any of the sublists of
↪   the second list.
```

```
=== assistant ===
Assistant:
<thought> I apologize for the syntax error. I should remove the
↪   backslashes from the function name and check if the elements of the
↪   first list are in any of the sublists of the second list. </thought>
<execute>
def check\_subset\_list(list1, list2):
    for sublist in list2:
        for element in list1:
            if element not in sublist:
                return False
    return True
res = check\_subset\_list(
    [1, 2, 3, 4, 5, 6, 7, 8, 9, 10, 11, 12, 13, 14],
    [[12, 18, 23, 25, 45],
    [7, 11, 19, 24, 28],
    [1, 5, 8, 18, 15, 16]]
)
assert res == False, "Expected False but got {}".format(res)
</execute>
```

```
=== user ===
Observation:
Cell In[2], line 1
    def check\_subset\_list(list1, list2):
                                            ^
SyntaxError: unexpected character after line continuation character
You have 3 steps left and 2 chances to propose solution left.
```

```
Expert feedback:
This is BAD. You have repeated the same syntax error in your function
↪   definition. The backslashes are not needed in the function name.
↪   Also, your logic is still incorrect. You are returning False as soon
↪   as you find an element from list1 that is not in a sublist of list2.
↪   This would mean that all elements of list1 need to be in all sublists
↪   of list2 for the function to return True. However, you should be
↪   checking if each element of list1 is in any sublist of list2.
```

```
=== assistant ===
Assistant:
<thought> I apologize for the repeated syntax error. I should remove the
↪   backslashes from the function name and check if each element of the
↪   first list is in any sublist of the second list. </thought>
<execute>
```

```
def check\_subset\_list(list1, list2):
    for element in list1:
        if element not in any(sublist for sublist in list2):
            return False
    return True
res = check\_subset\_list(
    [1, 2, 3, 4, 5, 6, 7, 8, 9, 10, 11, 12, 13, 14],
    [[12, 18, 23, 25, 45],
    [7, 11, 19, 24, 28],
    [1, 5, 8, 18, 15, 16]]
)
assert res == False, "Expected False but got {}".format(res)
</execute>
```

```
=== user ===
Observation:
Cell In[3], line 1
    def check\_subset\_list(list1, list2):
                                         ^

SyntaxError: unexpected character after line continuation character
You have 2 steps left and 2 chances to propose solution left.

Expert feedback:
This is BAD. You have repeated the same syntax error in your function
↪   definition. The backslashes are not needed in the function name.
↪   Also, your logic is still incorrect. You are using the 'any' function
↪   incorrectly. The 'any' function returns True if at least one element
↪   of an iterable is true. You should be using it to check if the
↪   element is in any of the sublists. You need to iterate over each
↪   sublist and check if the element is in the sublist. If the element is
↪   not in any of the sublists, then you should return False. If you have
↪   checked all elements and have not returned False, then you should
↪   return True.
```

```
=== assistant ===
Assistant:
<thought> I apologize for the repeated syntax error. I should remove the
↪   backslashes from the function name and use the 'any' function
↪   correctly to check if each element of the first list is in any
↪   sublist of the second list. </thought>
<execute>
def check\_subset\_list(list1, list2):
    for element in list1:
        if not any(element in sublist for sublist in list2):
            return False
    return True
res = check\_subset\_list(
    [1, 2, 3, 4, 5, 6, 7, 8, 9, 10, 11, 12, 13, 14],
    [[12, 18, 23, 25, 45],
    [7, 11, 19, 24, 28],
    [1, 5, 8, 18, 15, 16]]
)
assert res == False, "Expected False but got {}".format(res)
</execute>
```

```
=== user ===
Observation:
Cell In[4], line 1
    def check\_subset\_list(list1, list2):
                                         ^

SyntaxError: unexpected character after line continuation character
You have 1 steps left and 2 chances to propose solution left.
You should take the last step to propose a solution.

Expert feedback:
```

```
This is BAD. You have repeated the same syntax error in your function
↪  definition. The backslashes are not needed in the function name.
↪  Also, your logic is still incorrect. You are using the 'any' function
↪  incorrectly. The 'any' function returns True if at least one element
↪  of an iterable is true. You should be using it to check if the
↪  element is in any of the sublists. You need to iterate over each
↪  sublist and check if the element is in the sublist. If the element is
↪  not in any of the sublists, then you should return False. If you have
↪  checked all elements and have not returned False, then you should
↪  return True.

=== assistant ===
Assistant:
<solution>
The correct solution is:
```python
def check_subset_list(list1, list2):
    for element in list1:
        if not any(element in sublist for sublist in list2):
            return False
    return True
```

Explanation:

* We iterate over each element in `list1`.
* For each element, we check if it is in any of the sublists of `list2`
↪  using the `any()` function.
* If the element is not in any of the sublists, we return `False`.
* If we have checked all elements in `list1` and have not returned
↪  `False`, we return `True`.

This solution correctly checks if each element of `list1` is in any
↪  sublist of `list2`.
```

### E.2   CODELLAMA-INSTRUCT PRODUCES [PYTHON] TAG

The following is a random trajectory on coding tasks (in-context example omitted) from `CodeLLaMA-34b-Instruct` model[8]. Despite being explicitly instructed to output code within pairs of `<execute>` or `<solution>` tag through instruction and in-context example, `CodeLLaMA-34b-Instruct` *always* wraps their output solution within a pair of `[PYTHON]` tag as shown in Tab. A.10, despite not explicitly instructed so.

In Tab. A.10, we find that `[PYTHON]` tag is *always associated with code generation problems* but not other types of tasks. The appearance of `[PYTHON]` is most frequent in the largest version (34B), but not appearing at all in the smallest versions (7B).

We suspect the association of `[PYTHON]` tag and code generation task is due to artifacts introduced during the instruction-tuning phase of `CodeLLaMA-Instruct` model. Quoting Rozière et al. (2023), "`Code Llama - Instruct` variants are further fine-tuned on a mix of proprietary instruction data for improved safety and helpfulness, and **a new machine-generated self-instruct dataset created by prompting Llama 2 for coding problems** and Code Llama to generate associated unit tests and solutions." Furthermore, Rozière et al. (2023) mentioned that in the self-instruct process, they "**prompt the model to generate the code between [PYTHON] and [/PYTHON]** tags to make it easy to parse the result." We suspect `CodeLLaMA-Instruct` was explicitly fine-tuned on `[PYTHON]` tags during instruction-tuning, causing it to ignore instructions about other tags (e.g., `<execute>`) on coding problems. Furthermore, we hypothesize that the larger the LLM, the more it will be able to memorize data points seen during its instruction-tuning phase, causing it to produce `[PYTHON]` more often as observed in Tab. A.10.

---

[8] https://huggingface.co/CodeLLaMA/CodeLLaMA-34b-Instruct-hf

Table A.10: Percentage of task instances that assistant outputs [PYTHON] tag at least once, under the setting of $k = 5$ without language feedback. This seems to be a particular issue with CodeLLaMA LLMs on code-related tasks, especially the model with a larger scale.

| Evaluated LLM | Size | Type | Reasoning | Decision-Making | Code |
|---|---|---|---|---|---|
| **Percentage of assistant interactions that contains [PYTHON] tag (%)** | | | | | |
| **Open-source LLM** | | | | | |
| CodeLLaMA | 7B | Base | 0.00 | 0.00 | 0.00 |
| | | SIFT | 0.00 | 0.00 | 0.00 |
| | 13B | Base | 0.00 | 0.00 | 0.00 |
| | | SIFT | 0.00 | 0.00 | **2.21** |
| | 34B | Base | 0.00 | 0.00 | 0.00 |
| | | SIFT | 0.00 | 0.00 | **100.00** |
| LLaMA-2 | 7B | Base | 0.00 | 0.00 | 0.00 |
| | | RLHF | 0.00 | 0.00 | 0.00 |
| | 13B | Base | 0.00 | 0.00 | 0.00 |
| | | RLHF | 0.00 | 0.00 | 0.00 |
| | 70B | Base | 0.00 | 0.00 | 0.00 |
| Lemur-v1 | 70B | Base | 0.00 | 0.00 | 0.00 |
| | | SIFT | 0.00 | 0.00 | 0.00 |
| Vicuna-v1.5 | 7B | SIFT | 0.00 | 0.00 | 0.00 |
| | 13B | SIFT | 0.00 | 0.00 | 0.00 |
| **Closed-source LLM** | | | | | |
| chat-bison-001 | - | - | 0.00 | 0.00 | 0.00 |
| claude-2 | - | - | 0.00 | 0.00 | 0.00 |
| claude-instant-1 | - | - | 0.00 | 0.00 | 0.00 |
| gpt-3.5-turbo-0613 | - | - | 0.00 | 0.00 | 0.00 |
| gpt-3.5-turbo-16k-0613 | - | - | 0.00 | 0.00 | 0.00 |
| gpt-4-0613 | - | - | 0.00 | 0.00 | 0.00 |

```
=== user ===
You are a helpful assistant assigned with the task of problem-solving. To
↪  achieve this, you will be using an interactive coding environment
↪  equipped with a variety of tool functions to assist you throughout
↪  the process.

At each turn, you should first provide your step-by-step thinking for
↪  solving the task. Your thought process should be enclosed using
↪  "<thought>" tag, for example: <thought> I need to print "Hello
↪  World!" </thought>.

After that, you have two options:

1) Interact with a Python programming environment and receive the
↪  corresponding output. Your code should be enclosed using "<execute>"
↪  tag, for example: <execute> print("Hello World!") </execute>.
2) Directly provide a solution that adheres to the required format for
↪  the given task. Your solution should be enclosed using "<solution>"
↪  tag, for example: The answer is <solution> A </solution>.

You have 5 chances to interact with the environment or propose a
↪  solution. You can only propose a solution 2 times.

---

Task:
```

```
Complete the following code:

def is_nested(string):
    '''
    Create a function that takes a string as input which contains only
    ↪   square brackets.
    The function should return True if and only if there is a valid
    ↪   subsequence of brackets
    where at least one bracket in the subsequence is nested.

    is_nested('[[]]') -> True
    is_nested('[]]]]]]][[[[[]') -> False
    is_nested('[][]') -> False
    is_nested('[]') -> False
    is_nested('[[][]]') -> True
    is_nested('[[]][[') -> True
    '''

=== assistant ===
Assistant:

[PYTHON]
def is_nested(string):
    stack = []
    for char in string:
        if char == '[':
            stack.append(char)
        elif char == ']':
            if len(stack) == 0:
                return False
            stack.pop()
    return len(stack) == 0
[/PYTHON]

=== user ===
Observation:
I don't understand your input.
If you want to execute code, please use <execute> YOUR_CODE_HERE
↪   </execute>.
If you want to give me an answer, please use <solution>
↪   YOUR_SOLUTION_HERE </solution>.
For example: The answer to the question is <solution> 42 </solution>.

You have 4 steps left and 2 chances to propose solution left.

Expert feedback:
The assistant's approach to solving the problem is correct, but the
↪   assistant didn't follow the correct format for code execution. The
↪   assistant should use the <execute> tag to execute the code. Also, the
↪   assistant's solution doesn't seem to check for nested brackets, it
↪   only checks for balanced brackets. The assistant needs to modify the
↪   solution to check for nested brackets.

=== assistant ===
Assistant:

[PYTHON]
def is_nested(string):
    stack = []
    for char in string:
        if char == '[':
            stack.append(char)
        elif char == ']':
            if len(stack) == 0:
                return False
```

```
            stack.pop()
    return len(stack) == 0
[/PYTHON]

You have 4 steps left and 2 chances to propose solution left.

=== user ===
Observation:
I don't understand your input.
If you want to execute code, please use <execute> YOUR_CODE_HERE
↪   </execute>.
If you want to give me an answer, please use <solution>
↪   YOUR_SOLUTION_HERE </solution>.
For example: The answer to the question is <solution> 42 </solution>.

You have 3 steps left and 2 chances to propose solution left.

Expert feedback:
The assistant's approach to solving the problem is correct, but the
↪   assistant didn't follow the correct format for code execution. The
↪   assistant should use the <execute> tag to execute the code. Also, the
↪   assistant's solution doesn't seem to check for nested brackets, it
↪   only checks for balanced brackets. The assistant needs to modify the
↪   solution to check for nested brackets.

=== assistant ===
 Assistant:

I apologize for the confusion. I will make sure to follow the correct
↪   format for code execution and check for nested brackets in the
↪   solution.

Here is the modified solution:

<execute>
def is_nested(string):
    stack = []
    for char in string:
        if char == '[':
            stack.append(char)
        elif char == ']':
            if len(stack) == 0:
                return False
            stack.pop()
    return len(stack) == 0
</execute>

You have 3 steps left and 2 chances to propose solution left.
```

## F    PROMPTS AND EXAMPLES

We carefully designed prompts and in-context examples for each task type to elicit interaction be-
havior from LLMs. In this section, we include tool descriptions, error messages shown to LLM for
formatting issues, example task trajectories, prompt templates, and in-context examples.

### F.1    TOOL DESCRIPTIONS FOR EACH TASK TYPE

**For Code Generation**    No additional tool is provided for code generation apart from the Python
interpreter (§2.1).

**For Reasoning**

```
Tool function available (already imported in <execute> environment):
[1] wikipedia_search(query: str) -> str
The Wikipedia Search tool provides access to a vast collection of
↪   articles covering a wide range of topics.
Can query specific keywords or topics to retrieve accurate and
↪   comprehensive information.
```

**For Decision-Making (ALFWorld)**

```
Tool function available (already imported in <execute> environment):
[1] put(object: str, receptacle: str) -> str
Put an object in/on a receptacle.
For example: put("mug 1", "desk 2")

[2] goto(receptacle: str) -> str
Go to a location of the receptacle.
For example: goto("drawer 1")

[3] take_from(object: str, receptacle: str) -> str
Take an object from a receptacle.
For example: take_from("mug 1", "shelf 2")

[4] open_receptacle(receptacle: str) -> str
Open a receptacle.
For example: open_receptacle("fridge 1")

[5] toggle(object_or_receptacle: str) -> str
Toggle an object or receptacle.
For example: toggle("light 2")

[6] close_receptacle(receptacle: str) -> str
Close a receptacle.
For example: close_receptacle("microwave 1")

[7] clean(object: str, receptacle: str) -> str
Clean an object with a receptacle.
For example: clean("cloth 1", "sinkbasin 1")

[8] heat(object: str, receptacle: str) -> str
Heat an object with a receptacle.
For example: heat("egg 1", "microwave 1")

[9] cool(object: str, receptacle: str) -> str
Cool an object with a receptacle.
For example: cool("bottle 1", "fridge 1")

[10] use(receptacle: str) -> str
Use a receptacle.
For example: use("lamp 1")

[11] look() -> str
Look around. It will return what you see in the room.
For example: look()
```

## F.2   FORMATTING ERROR MESSAGE

The following error message is provided to the LLM Agent when we fail to detect the instructed format (e.g., fail to find `<execute>` and/or `<solution>` tag).

```
I don't understand your input.
If you want to execute code, please use <execute> YOUR_CODE_HERE
↪   </execute>.
```

```
If you want to give me an answer, please use <solution>
↪   YOUR_SOLUTION_HERE </solution>.
For example: The answer to the question is <solution> 42 </solution>.
```

## F.3 EXAMPLE TASK TRAJECTORY

The following is an example trajectory on a reasoning task from MATH (Hendrycks et al., 2021), generated by `gpt-3.5-turbo-0613` as an agent and `gpt-4-0613` as a simulated human advisor to provide feedback. We release outputs from all LLMs we evaluated in our code repo.

```
Task:
Let $a,$ $b,$ and $c$ be the roots of $x^3 - 3x - 5 = 0.$  Find $abc.$

Assistant:
<thought> To solve this problem, we can use Vieta's formulas. Vieta's
↪   formulas state that for a polynomial $ax^n + bx^{n-1} + cx^{n-2} +
↪   \dots + k = 0$ with roots $r_1, r_2, \dots, r_n$, the following
↪   relationships hold:

$r_1 + r_2 + \dots + r_n = -\frac{b}{a}$
$r_1r_2 + r_1r_3 + \dots + r_{n-1}r_n = \frac{c}{a}$
$r_1r_2r_3 + r_1r_2r_4 + \dots + r_{n-2}r_{n-1}r_n = -\frac{d}{a}$
$\dots$
$r_1r_2 \dots r_n = (-1)^n \frac{k}{a}$

In this case, we have a cubic polynomial $x^3 - 3x - 5 = 0$, so we can
↪   use the formula $abc = -\frac{d}{a}$ to find the product of the
↪   roots. </thought>
<execute>
a = 1
d = 5
abc = -d/a
abc
</execute>

Observation:
Out[1]: -5.0
You have 4 steps left and 2 chances to propose solution left.

Expert feedback:
This is GOOD. You correctly applied Vieta's formulas to find the product
↪   of the roots of the given cubic equation. Now you can submit the
↪   solution.

Assistant:
The answer is <solution> -5.0 </solution>.

Observation:
Your answer is wrong.
You have 3 steps left and 1 chances to propose solution left.

Expert feedback:
The assistant made a mistake in applying Vieta's formulas. The formula
↪   for the product of the roots of a cubic equation is $abc = (-1)^n
↪   \frac{k}{a}$, not $abc = -\frac{d}{a}$. In this case, $n=3$, $k=-5$,
↪   and $a=1$, so the correct formula to use is $abc = (-1)^3
↪   \frac{-5}{1}$. Please correct this and try again.

Assistant:
<thought> I apologize for the mistake. Let's correct it. </thought>
<execute>
a = 1
k = -5
abc = (-1)**3 * k/a
</execute>
```

```
abc
</execute>

Observation:
Out[2]: 5.0
You have 2 steps left and 1 chances to propose solution left.

Expert feedback:
This is GOOD. You have correctly applied Vieta's formulas this time and
↪  found the correct product of the roots of the given cubic equation.
↪  Now you can submit the solution.

Assistant:
The answer is <solution> 5.0 </solution>.
```

## F.4   PROMPT TEMPLATES

To instantiate the prompt for each task type, {in_context_example} will be substituted with examples in §F.5, and {tool_desc} will be substituted with tool descriptions in §F.1.

### F.4.1   EVALUATE LLM

For each task, we replace {task_prompt} with the corresponding task instruction. {max_total_steps} and {max_propose_steps} will be substitute with the setting for that experiments. In MINT, we default to 5 max total steps and 2 max propose steps as discussed in §2.1 and §3.1.

```
You are a helpful assistant assigned with the task of problem-solving. To
↪  achieve this, you will be using an interactive coding environment
↪  equipped with a variety of tool functions to assist you throughout
↪  the process.

At each turn, you should first provide your step-by-step thinking for
↪  solving the task. Your thought process should be enclosed using
↪  "<thought>" tag, for example: <thought> I need to print "Hello
↪  World!" </thought>.

After that, you have two options:

1) Interact with a Python programming environment and receive the
↪  corresponding output. Your code should be enclosed using "<execute>"
↪  tag, for example: <execute> print("Hello World!") </execute>.
2) Directly provide a solution that adheres to the required format for
↪  the given task. Your solution should be enclosed using "<solution>"
↪  tag, for example: The answer is <solution> A </solution>.

You have {max_total_steps} chances to interact with the environment or
↪  propose a solution. You can only propose a solution
↪  {max_propose_solution} times.

{tool_desc}

---

{in_context_example}

---

{task_prompt}
```

### F.4.2 SIMULATE LANGUAGE FEEDBACK

To instantiate the template for feedback generation, we will replace {trajectory} with an LLM agent's trajectory (e.g., §F.3). When the ground-truth solution is not provided for feedback generation, {gt_solution} will be substituted with "NOT GIVEN"; Otherwise, the ground-truth solution for that task will be provided.

```
You are an expert tasked with evaluating and providing feedback on an
↪  assistant's performance.

---
Here is an example. Please follow the format as the following expert
↪  acts.

{in_context_example}

---

{tool_desc}

{trajectory}

Correct solution (please DO NOT disclose the correct solution to the
↪  assistant): {gt_solution}

Please provide concise and constructive feedback. Remember, your role is
↪  similar to a teacher. Rather than giving away the solution or details
↪  about the answer, guide the assistant toward understanding how to
↪  arrive at the correct answer. Your feedback should focus on enhancing
↪  the assistant's ability to think critically and respond accurately.
↪  Now provide your feedback.
Expert feedback:
```

### F.5 IN-CONTEXT EXAMPLE

We perform 1-shot learning on all tasks in MINT. We have two types of in-context examples for each task: (1) without feedback, and (2) with feedback. We show the in-context examples with the provided feedback below. When no feedback is provided to the LLM (i.e., without feedback in-context examples), we will remove all occurrences of "Expert feedback: ..." in the following examples.

We curate one in-context example for code generation and reasoning problems, and we follow Yao et al. (2022) to construct examples for each sub-type of decision-making (i.e., ALFWorld) tasks. For each task, we include one corresponding example for in-context learning. For simplicity, we only include a subset of prompts here. All prompts are released in our code repo.

### F.5.1 DECISION-MAKING

We modify in-context prompts from Yao et al. (2022) to construct these in-context examples. This is an example of clean type of tasks in ALFWorld.

```
Task:
You are in the middle of a room. Looking quickly around you, you see a
↪  cabinet 13, a cabinet 12, a cabinet 11, a cabinet 10, a cabinet 9, a
↪  cabinet 8, a cabinet 7, a cabinet 6, a cabinet 5, a cabinet 4, a
↪  cabinet 3, a cabinet 2, a cabinet 1, a coffeemachine 1, a countertop
↪  1, a diningtable 1, a drawer 1, a fridge 1, a garbagecan 1, a
↪  microwave 1, a shelf 3, a shelf 2, a shelf 1, a sinkbasin 1, a
↪  stoveburner 4, a stoveburner 3, a stoveburner 2, a stoveburner 1, and
↪  a toaster 1.
Your task is to: put a clean lettuce in diningtable.

Assistant:
```

```
<thought>
To solve the task, I need to find a lettuce, then clean it with
↪  sinkbasin, then put it in diningtable.
First I need to find a lettuce. A lettuce is more likely to appear in
↪  fridge (1), diningtable (1), sinkbasin (1), stoveburner (1-3),
↪  cabinet (1-13). I can check one by one.
</thought>
<execute>
for loc_to_check in ["fridge 1", "diningtable 1",
"sinkbasin 1", "stoveburner 1",
"stoveburner 2", "stoveburner 3",
*[f"cabinet {i}" for i in range(1, 14)]]:
    res = goto(loc_to_check)
    if "closed" in res:
        # The receptacle is closed. I need to open it to see what's
        ↪  inside.
        res = open_receptacle(loc_to_check)

    if "lettuce" in res:
        print(res)
        print(f"Found a lettuce in {loc_to_check}.")
        break
</execute>

Observation:
On the diningtable 1, you see a apple 1, a bread 1, a butterknife 2, a
↪  cup 2, a fork 2, a knife 2, a knife 1, a ladle 1, a lettuce 1, a mug
↪  2, a mug 1, a pan 2, a peppershaker 1, a spatula 3, a tomato 2, and a
↪  tomato 1.
Found a lettuce in diningtable 1.

Assistant:
<thought>
Now I find a lettuce 1. Next, I need to take it and go to sinkbasin 1 and
↪  clean the lettuce 1.
Once it is cleaned, I need to put it in diningtable 1.
</thought>
<execute>
print(take_from("lettuce 1", "diningtable 1"))
print(goto("sinkbasin 1"))
print(clean("lettuce 1", "sinkbasin 1"))
print(goto("diningtable 1"))
print(put("lettuce 1", "diningtable 1"))
</execute>

Observation:
You pick up the lettuce 1 from the diningtable 1.
On the sinkbasin 1, you see a apple 2, a ladle 2, a spoon 1, and a tomato
↪  3.
You clean the lettuce 1 using the sinkbasin 1.
On the diningtable 1, you see a apple 1, a bread 1, a butterknife 2, a
↪  cup 2, a fork 2, a knife 2, a knife 1, a ladle 1, a mug 2, a mug 1, a
↪  pan 2, a peppershaker 1, a spatula 3, a tomato 2, and a tomato 1.
You put the lettuce 1 in/on the diningtable 1.

Assistant:
<thought> I have completed the task. </thought>
<solution> done </solution>
```

### F.5.2 CODE GENERATION

We select one example from MBPP's in-context example subset (Austin et al., 2021) and curate it to fit MINT framework (i.e., adding observations and feedback). We use the same in-context example

after a slight format adjustment for HumanEval (Chen et al., 2021) since there were no dedicated in-context examples provided for HumanEval.

```
Task:
Write a function to find the similar elements from the given two tuple
↪   lists.
assert similar_elements((3, 4, 5, 6),(5, 7, 4, 10)) == (4, 5)

Assistant:
<thought> The similar elements here are just elements that are in both
↪   lists. I should run a solution on the test case before proposing a
↪   solution. </thought>
<execute>
def similar_elements(test_tup1, test_tup2):
    res = tuple(set(test_tup1) | set(test_tup2))
    return res
res = similar_elements((3, 4, 5, 6), (5, 7, 4, 10))
assert res == (4, 5), "Expected (4, 5) but got {}".format(res)
</execute>

Observation:
---------------------------------------------------------------------
AssertionError                              Traceback (most recent call
↪   last)
Cell In[4], line 5
      3     return res
      4 res = similar_elements((3, 4, 5, 6), (5, 7, 4, 10))
----> 5 assert res == (4, 5), "Expected (4, 5) but got {}".format(res)

AssertionError: Expected (4, 5) but got (3, 4, 5, 6, 7, 10)

Expert feedback:
This is BAD. You should not use the | operator which is the union
↪   operator.

Assistant:
<thought> To find the similar elements, I should find the intersection of
↪   the two sets. I should use the & operator instead of the | operator.
↪   </thought>
<execute>
def similar_elements(test_tup1, test_tup2):
    res = tuple(set(test_tup1) & set(test_tup2))
    return res
res = similar_elements((3, 4, 5, 6), (5, 7, 4, 10))
assert res == (4, 5), "Expected (4, 5) but got {}".format(res)
</execute>

Observation:
[Executed Successfully with No Output]

Expert feedback:
This is GOOD. You are doing great! You can try to submit the solution
↪   now.

Assistant:
<thought> There is no more AssertionError. I can now submit the solution.
↪   </thought>
<solution>
def similar_elements(test_tup1, test_tup2):
    res = tuple(set(test_tup1) & set(test_tup2))
    return res
</solution>
```

### F.5.3 REASONING TASK

We manually curate the in-context example for the reasoning subset to include the characteristic for both multi-hop QA-style reasoning (e.g., HotpotQA) and mathematical reasoning (e.g., MATH, TheoremQA).

```
Task:
Evaluate $\int\int_C (x+y)^2 dx,dy$, where C is the circle defined by
↪  $x^2 + y^2 = M$, and M is the birth year of Li Bai.
Output format requirement: Round to three decimal places, using half up
↪  rounding rule.

Assistant:
<thought> To solve this problem, let's start by determining Li Bai's
↪  birth year. </thought>
<execute>
print(wikipedia_search('Li Bai'))
</execute>

Observation:
Page: Li Bai
Summary: Li Bai (pinyin: Lǐ Bái, 701{762), also pronounced as Li Bo,
↪  courtesy name Taibai, was a Chinese poet, acclaimed from his own time
↪  to the present as a brilliant and romantic figure who took
↪  traditional poetic forms to new heights.

Expert feedback:
This is GOOD. You have successfully retrieved the birth year of Li Bai
↪  from the wiki search result. Next you should try to solve the
↪  calculus problem with the information you obtained.

Assistant:
<thought>
From the wiki_search result, we can see that Li Bai was born in 701, thus
↪  M = 701.
To evaluate the double integral of $(x+y)^2$ over the region $C$, where
↪  $C$ is the circle defined by $x^2 + y^2 = 701$, we should use polar
↪  coordinates.
In polar coordinates, $x = r\cos(\theta)$ and $y = r\sin(\theta)$, and
↪  the differential area element $dx,dy$ becomes $r,dr,d\theta$.
$\int\int_C (x+y)^2 dx,dy = \int_0^{2\pi}\int_0^{\sqrt{701}}
↪  (r\cos(\theta) + r\sin(\theta))^2r,dr,d\theta$
Let's use the SymPy package to calculate this.
</thought>
<execute>
from sympy import symbols, integrate, cos, sin, pi

r, theta = symbols('r theta')

# the integrand
integrand = (r*cos(theta) + r*sin(theta))**2 * r

# perform the double integration
double_integral = integrate(integrate(
    integrand, (r, 0, 701 ** 0.5)),
    (theta, 0, 2*pi)
)
print(double_integral)
</execute>

Observation:
Out[2]: 245700.5*pi

Expert feedback:
```

```
This is GOOD. You successfully converted the double integral into a form
↪  that can be calculated. Now you can convert the answer to meet the
↪  output format requirement.

Assistant:
<thought> Let's round this result to three decimal places using half up
↪  rounding rule. </thought>
<execute>
import math
print(round(245700.5*math.pi, 3))
</execute>

Observation:
771890.886

Expert feedback:
This is GOOD. You have got the solution! You can try to submit the
↪  solution now.

Assistant:
The answer is <solution> 771890.886 </solution>.
```

## F.6 EXAMPLE GROUND-TRUTH FOR DECISION-MAKING TASK

The following is an example ground-truth solution for a task from Alfworld Shridhar et al. (2020).

```
Task:
Interact with a household to solve a task.
You are in the middle of a room. Looking quickly around you, you see a
↪  cabinet 6, a cabinet 5, a cabinet 4, a cabinet 3, a cabinet 2, a
↪  cabinet 1, a coffeemachine 1, a countertop 3, a countertop 2, a
↪  countertop 1, a drawer 3, a drawer 2, a drawer 1, a fridge 1, a
↪  garbagecan 1, a microwave 1, a shelf 3, a shelf 2, a shelf 1, a
↪  sinkbasin 1, a stoveburner 4, a stoveburner 3, a stoveburner 2, a
↪  stoveburner 1, and a toaster 1.
Your task is to: put two peppershaker in drawer.

Assistant:
<execute>
goto("shelf 1")
goto("shelf 2")
goto("shelf 3")
goto("drawer 1")
</execute>

Observation:
On the shelf 1, you see nothing.
On the shelf 2, you see a bowl 1, and a saltshaker 2.
On the shelf 3, you see a vase 2.
The drawer 1 is closed.

Assistant:
<execute>
open("drawer 1")
close("drawer 1")
goto("drawer 2")
open("drawer 2")
close("drawer 2")
goto("drawer 3")
open("drawer 3")
close("drawer 3")
goto("countertop 1")
goto("countertop 2")
take("peppershaker 3", "countertop 2")
```

```
goto("drawer 1")
open("drawer 1")
put("peppershaker 3", "drawer 1")
close("drawer 1")
goto("shelf 1")
goto("shelf 2")
goto("shelf 3")
goto("drawer 2")
open("drawer 2")
close("drawer 2")
goto("drawer 3")
open("drawer 3")
close("drawer 3")
goto("countertop 1")
goto("countertop 2")
goto("countertop 3")
goto("cabinet 1")
take("peppershaker 2", "cabinet 1")
goto("drawer 1")
open("drawer 1")
put("peppershaker 2", "drawer 1")
</execute>

Observation:
You open the drawer 1. The drawer 1 is open. In it, you see a cellphone
↪  1, and a fork 1.
You close the drawer 1.
The drawer 2 is closed.
You open the drawer 2. The drawer 2 is open. In it, you see nothing.
You close the drawer 2.
The drawer 3 is closed.
You open the drawer 3. The drawer 3 is open. In it, you see a spatula 2.
You close the drawer 3.
On the countertop 1, you see a cup 1, a fork 2, a mug 3, a soapbottle 3,
↪  a soapbottle 2, a spatula 1, and a spoon 1.
On the countertop 2, you see a butterknife 1, a knife 1, a mug 1, a
↪  peppershaker 3, and a tomato 1.
You pick up the peppershaker 3 from the countertop 2.
The drawer 1 is closed.
You open the drawer 1. The drawer 1 is open. In it, you see a cellphone
↪  1, and a fork 1.
You put the peppershaker 3 in/on the drawer 1.
You close the drawer 1.
On the shelf 1, you see nothing.
On the shelf 2, you see a bowl 1, and a saltshaker 2.
On the shelf 3, you see a vase 2.
The drawer 2 is closed.
You open the drawer 2. The drawer 2 is open. In it, you see nothing.
You close the drawer 2.
The drawer 3 is closed.
You open the drawer 3. The drawer 3 is open. In it, you see a spatula 2.
You close the drawer 3.
On the countertop 1, you see a cup 1, a fork 2, a mug 3, a soapbottle 3,
↪  a soapbottle 2, a spatula 1, and a spoon 1.
On the countertop 2, you see a butterknife 1, a knife 1, a mug 1, and a
↪  tomato 1.
On the countertop 3, you see a apple 2, a bread 1, a cellphone 3, a
↪  cellphone 2, a creditcard 1, a glassbottle 2, a houseplant 1, a plate
↪  2, a pot 2, a spatula 3, a spoon 3, a spoon 2, and a statue 1.
On the cabinet 1, you see a mug 2, and a peppershaker 2.
You pick up the peppershaker 2 from the cabinet 1.
The drawer 1 is closed.
You open the drawer 1. The drawer 1 is open. In it, you see a cellphone
↪  1, a fork 1, and a peppershaker 3.
You put the peppershaker 2 in/on the drawer 1.
```

