# OpenReview forum: "MINT: Evaluating LLMs in Multi-turn Interaction with Tools and Language Feedback"
_ICLR.cc/2024/Conference — ICLR 2024 poster_

### Official Review · Reviewer_Nxjw · 2023-10-31

**Soundness:** 3 good
**Presentation:** 3 good
**Contribution:** 3 good
**Rating:** 5
**Confidence:** 5

**Summary:**

This paper evaluates LLMs in multi-turn interaction with the help of tools and language feedback. The authors introduces the MINT benchmark composing of code generation, decision making, and reasoning tasks, and compared 20 open- and closed-source LLMs. Results show that LLMs benefit from multiple rounds of tool-use and language feedback, and LLMs trained with supervised instruction-finetuning (SIFT) and RLHF may hurt multi-turn performance. Moreover, better single-turn performance does not generalize to better multi-turn performance.

**Strengths:**

1. This paper proposes an interesting MINT benchmark to evaluate multi-turn interactions using code execution and language feedbacks. The modification of using a unified subset of examples across a bundle of tasks would be useful for the community in general.
2. Results showing that supervised instruction-finetuning and RLHF may hurt model performance of multi-turn interactions with tools and language feedback, and worse performance on multi-turn compared to single-turn, may suggest that multi-turn interaction data is required. This provides valuable suggestions to alignment research.

**Weaknesses:**

1. There is no comparison to simple baselines, such as self-critic (which is essentially a special case of multi-turn interactions defined in this paper). Other baselines include only providing binary feedback, or sample k times and ask a LLM to select the final answer. Without such baselines, it is not clear how much language feedback the model actually incorporates in. Furthermore, I generally like the idea t hat using a LLM to provide feedback can improve a model's performance in general, but can the authors provide any benefit (such as efficiency) comparing using a large LLM as a feedback provider, compared to using a LLM to generate the results directly (either using self-critic or not)?
2. The proposed MINT dataset is relatively biased towards math problems that specifically require code execution and can benefit from explicit human feedback (e.g., correct or incorrect).  It seems that the only tool used is the python execution tool. Compared to other papers leveraging tools (toolformer, ReAct), the claim of utilizing tools through turns would improve model performance is not very convincing, especially for tasks that may not require code execution (e.g., HotpotQA). Moreover, I agree that selecting a subset of datasets would hinder fair comparisons, but sampling for instances, only 43 examples for HotpotQA (which is arguably the only multihop-qa dataset in MINT) may not be statistically large enough to represent the performance on the dataset. More importantly, MINT is limited to multi-turn interactions using the specific code tools and language feedback used in this paper, not the more natural and systematic tasks that require multi-turn interactions.
3. Accordingly, results suggesting that for example, "better single-turn performance does not guarantee better multi-turn performance" and "RLHG may hurt multi-turn capabilities" are limited to the setup that those worse performance would largely be due to "less viable of incorporating feedback" rather than necessarily indicating that they are indeed worse on multi-turn interactions with agents in general. Therefore, the claims are not well justified.

**Questions:**

1. Where did you compare to the "Lazy User-LLM Interaction"? Or is this only for k=1?
2. The proposed method is very similar to the Tree-of-thought paper (Yao et al., 2023) where the authors used LMs as a feedback/reward provider.

Yao et al., 2023. Deliberate Problem Solving with Large Language Models.

---

> ### Author Response · Authors · 2023-11-19
>
> We thank the reviewer for their detailed responses, and we are glad that the reviewer find our paper important and easy to follow.
>
> > … no comparison to simple baselines, such as self-critic ...
>
> We evaluate self-critic in section 3.4 when LLMs provide feedback for themself. We find that even GPT-4 suffers from performance degradation, suggesting the limited performance gain from self-critic.
>
> > … can the authors provide any benefit (such as efficiency) comparing using a large LLM as a feedback provider, compared to using a LLM to generate the results directly…?
>
> We believe such a comparison is irrelevant to our research questions. MINT never intends to use a large model to *improve* the performance of the smaller model, but to **evaluate** the latter's capabilities.
>
> One of MINT’s goals is to **quantify a model’s ability to leverage language feedback**, which, in the real-world setting, is provided by users in their multi-turn interaction with LLM. The decision to employ GPT-4 (a larger/stronger LLM) as a feedback provider was driven by the need to establish a reproducible way to simulate users' language feedback for the evaluation of LLMs.
>
> > It seems that the only tool used is the python execution tool…
>
> Please refer to the general response.
>
> > … dataset … biased towards math problems that specifically require code execution and can benefit from explicit human feedback … not the more natural and systematic tasks … only 43 examples for HotpotQA … may not be statistically large enough to represent the performance…
>
> We argue that MINT is *not* biased toward math (~25% of instances are math). It covers a wide range of tasks, including math, code generation, decision-making, and multihop QA, that reflect the representative and commonly occurring tasks that can benefit from tool & feedback (our goal of evaluation).
>
> Regarding the size of our dataset, we had to trade-off between the number of examples and the cost of evaluation. Our original dataset contains 29k entries; while we could have included all these data points for evaluation with sufficient funding, it would have been prohibitively expensive (for GPT-4 to provide feedback) for the academic community to use and hindered the broader research and dissemination.
>
> In preliminary experiments, we found that the models' performance trend across the selected data points is consistent and representative of that on a larger selection. Our decision on downsampling largely reduces the cost of MINT without negatively impacting the conclusions one can draw with it. We believe that a sample of 500 entries that requires interaction is an appropriate compromise, as seen in other datasets in the field (e.g., TheoremQA, AlfWorld).
>
> We will also release the script we used to down-sample these datasets so users may choose to up-sample the HotpotQA subset when deemed necessary.
>
> > results … are limited to the setup that those worse performance would largely be due to "less viable of incorporating feedback" rather than necessarily indicating that they are indeed worse on multi-turn interactions with agents in general...
>
> We would like to clarify that these findings hold regardless of whether the GPT-4 simulated feedback is included, as discussed at the end of the introduction. Particularly, in Section 3.2, we find supporting evidence that even when **no GPT-4 feedback is provided at all, the claims the reviewer mentioned still hold**.
>
> If the reviewer is referring to “feedback” from the Python interpreter (i.e., observation), we believe that the capabilities of improving from *observations* in multi-turn interaction and incorporating natural language feedback are highly correlated. We are happy to hear the reviewers' thoughts on how to disentangle them and include experiments to verify them accordingly.
>
> > Where did you compare to the "Lazy User-LLM Interaction"? Or is this only for k=1?
>
> As explained in Section 2.1, “Lazy User-LLM Interaction” defines the behavior of users in interaction (i.e., provides *at most one response* when the solution is wrong). This alone corresponds to the setting of k=2 (2 turns of LLM response): LLM proposes answer (at most twice) for each response with no tools used, which deviates from our goal of evaluating tool and feedback incorporation capabilities. Therefore, “Lazy User-LLM Interaction” is just a description of the user behavior when no GPT-4 simulated feedback is involved. We will make this clearer in the next revision.
>
> > The proposed method is very similar to the Tree-of-thought paper (Yao et al., 2023) where the authors used LMs as a feedback/reward provider
>
> We emphasize that our focus **differs significantly** from that of Tree-of-thought (ToT). Despite some similarity in methodology, MINT focuses on **evaluating** any LLM under the setting of multi-turn interactions with tools and feedback, which diverge substantially from the objectives of the ToT that shows prompting LLM in ToT improves problem-solving performance.

---

> > ### Comment · Reviewer_Nxjw · 2023-11-22
> >
> > Thanks for the responses.
> >
> > I would like to clarify that I understand that the goal of the paper is to evaluate LLMs and there is a computational constraint. However, I am still not convinced by the claim of "interaction with tools" where python interpreter is the only tool evaluated. Furthermore, regarding the size of data, if sampling cost is a limit (to verify the statistic significance), you may be able to show the performance on a larger sample set using closed models. If the claim is to evaluate on different types of tasks, number of examples per category should be adequate.

---

> > > ### Author Response · Authors · 2023-11-23
> > >
> > > Thanks for the follow-up!
> > >
> > > > However, I am still not convinced by the claim of "interaction with tools" where python interpreter is the only tool evaluated.
> > >
> > > We argue that programming language function calls, such as Python function calls, are a superset of current tool using paradigms, which means that all tools can be easily converted to Python functions.
> > >
> > > For example, in ReAct, an LLM might call a search tool by generating:
> > >
> > > `Action: Search[Something]`
> > >
> > > This string is parsed by their [source code](https://github.com/ysymyth/ReAct/blob/6bdb3a1fd38b8188fc7ba4102969fe483df8fdc9/wikienv.py#L132-L137) to convert to an actual Python function call to Wikipedia.
> > >
> > > In MINT, we are given the tool as a function to LLM and expect it to *directly call the underlying function*. The benefit of this is that we can get rid of manual parsings for all tools with different formats (e.g., different numbers of arguments) and be able to provide Python interpreter error messages to the model directly when something goes wrong (e.g., formatting) similar to [1].
> > >
> > > `<execute> print(wikipedia_search(“Something”) </execute>`
> > >
> > > Beyond our explicitly given tools by our task-specific prompts (e.g., Wikipedia search in Table A.8). It's worth noting that we also consider implicit Python packages. An example is shown in Figure 1; the LLM is able to import relevant tools (i.e., python packages) when needed (e.g. to solve an equation):
> > >
> > > ```pytho
> > > <execute>
> > > from sympy import symbols, Eq, solve
> > > x, b = symbols(‘x b’)
> > > equation = Eq(x + (x+7) + (1/2) * b, 43)
> > > solve(equation, x)
> > > </execute>
> > > ```
> > >
> > > We want to emphasize that **Python Interpreter is not one single tool, but an interface to multiple tools** that cover tools in existing work (e.g., ReAct) and go beyond the provided tool set using the existing Python libraries (e.g., Python itself can act as a calculator, sympy package’s `solve` is an equation solving tool, etc.).
> > >
> > > Therefore, we think the Python Interpreter setting can show the tool use capabilities of current language models and align with our goal of evaluating the model's ability to use tools in diverse contexts. We will clarify this in the next revision.
> > >
> > > [1] Teaching Large Language Models to Self-Debug
> > >
> > > > Furthermore, regarding the size of data, if sampling cost is a limit (to verify the statistic significance), you may be able to show the performance on a larger sample set using closed models. If the claim is to evaluate on different types of tasks, number of examples per category should be adequate.
> > >
> > > It’s worth noting that MINT evaluates the model’s **multi-turn interaction performance** rather than single-turn problem-solving performance. MINT *repurposes* existing datasets by selecting examples in the original dataset that are challenging and *require interaction to solve* as discussed in Section 2.2.
> > >
> > > Therefore, in the scope of MINT, there is little meaning to evaluating models on the whole set, since not every example is challenging and requires multi-turn interaction. Below are the dataset statistics after filtering out non-interactive instances from the original dataset (Appendix Section D).
> > >
> > > | Category           | Dataset   | Count |
> > > |--------------------|-----------|-------|
> > > | Decision-Making           | Alfworld  | 134   |
> > > | Reasoning          | TheoremQA | 268   |
> > > | Reasoning          | GSM8K     | 240   |
> > > | Reasoning          | HotpotQA  | 221   |
> > > | Reasoning          | Math      | 1810  |
> > > | Reasoning          | MMLU      | 770   |
> > > | Code Generation    | MBPP      | 184   |
> > > | Code Generation    | HumanEval | 90    |
> > > | **Total**          |           | 3717  |
> > >
> > > On this filtered dataset, we perform stratified sampling and results in 586 instances, which we believe is representative of the filtered set. Note for Alfworld, the original dataset only has 134 test samples in total. Hence, we did not perform any additional filtering.
> > >
> > > We are not sure what the reviewer means by “closed models”; we assume that we are asked to evaluate open-source model performance on a larger set since they are relatively cheaper to run. In fact, the major costs of MINT come from the need for GPT-4 to simulate language feedback, which *scales up with the number of examples* regardless of the evaluated model, so running open-source model on more examples will not be cheaper. Furthermore, even if we do evaluate the open-source model on a larger set, these results cannot be fairly compared to those closed-source models on the sampled set. Therefore, we believe the current sample size is an adequate trade-off between quality and cost. We are happy to hear the reviewers' thoughts on how to empirically find the ideal number of samples required for each task without incurring too much cost to stay within our budget, and we can add such experiments in our next revision.

---

### Official Review · Reviewer_hgTu · 2023-10-31

**Soundness:** 4 excellent
**Presentation:** 2 fair
**Contribution:** 3 good
**Rating:** 8
**Confidence:** 4

**Summary:**

This paper presents MINT, a benchmark that evaluates LLMs’ multi-turn interaction capabilities, in particular when LLMs are using external tools and provided with natural language feedback from users (simulated by GPT-4). The experiments use a subset of existing evaluation datasets on reasoning, coding, and decision-making tasks, resulting in 586 instances, which require multi-turn interaction (judged by GPT-3.5).
The authors compare 20 open- and close-sourced LLMs on these instances and present 7 findings (listing them in order to easily refer to them in the other parts of the review):

1. All models benefit from tool interaction and natural language feedback
2. Better single-turn performance does not lead to better multi-turn performance
3. There is a notable gap between open- and closed-source LLMs in multi-turn interaction performance
4. Models trained with supervised instruction fine-tuning (SIFT) and reinforcement learning from human feedback (RLHF) perform worse in multi-turn settings

From the text:

5. LLMs’ feedback-providing capability could be orthogonal to task-solving ability
6. MINT’s evaluation reveals undesired artifacts in ShareGPT data
7. GPT-4 simulated language feedback is as helpful as human-written feedback based on human evaluation and task performance.

**Strengths:**

* The work studies a **timely, important topic** of LLMs’ evaluation in interactive settings, striving to bridge the gap between traditional static/single-turn benchmarks in NLP and “real-world use cases” (as noted in abstract). The use of LLMs as a proxy for human feedback reduces cost significantly and increases scalability.
* The paper is overall **well-written** and the design of the experiments seem **sound**. In addition, I find the **details of the experiments** are properly documented either in the main paper or appendix.
* Even if the exact numbers in the findings are likely to become less relevant in the future due to the fast-changing landscape of LLMs, the **high-level findings and takeaways are likely to be still relevant**, especially about single-turn vs. multi-turn performance between models (e.g., #2 and #4).

**Weaknesses:**

* Although the paper presents many interesting findings, it wasn’t clear to me which ones are **new findings vs. confirmation of existing findings**. For instance, #1 has been shown by ToolLLM (Qin et al., 2023), Tool Augmented Language Models (TALM) (Parisi et al., 2022), Chain of Hindsight (Liu et al., 2023), etc.;  and #2, #3, and #4 have been (fully or partially) shown by Human-AI Language-based Interaction Evaluation (HALIE) (Lee et al., 2023); and #7 has been studied by Liu et al. (2023), Chiang & Lee (2023), Gao et al (2023), Shen et al (2023), and Duan et al. (2023). Situating the findings in the existing literature will give this work more validity and help readers understand the contributions of the paper.
* For that reason, it is a bit unclear to me what the **main contribution of this paper** is at the moment. In comparison to prior work, the authors note that “Different from prior work, MINT covers a range of diverse tasks and is designed to measure the multi-turn interaction capabilities of LLMs with both tools and user feedback that are more aligned with real-world applications.” As some of the prior work covers a range of diverse tasks/measures multi-turn interaction capabilities of LLMs/simulates human feedback with LLMs, the main difference of this work is (presumably) accounting for **the use of tools and user feedback** on top of all of these elements. In this case, I wonder if it'll help to scope the description of the tasks more tightly (e.g., tasks that can benefit from tool use, as opposed to open-ended generation) and provide a bit more justification for why it’s important to look at these elements altogether (as there are prior work looking at the tool use and NL feedback) and what we expect to see (do we expect to see different results when these elements are added? do we actually observe any interesting interplay between these elements or is it merely a simple performance boost across all models?).

**Questions:**

* Most of the questions that can change my opinion are described in the weaknesses. I'm generally excited about the direction of the paper, and with clarification and contextualization of this work’s contributions and findings with respect to prior/concurrent work, I’d be happy to reconsider and adjust my ratings. Here, I list some questions/comments I had while reading the paper.
* Maybe another bit that’s quite important to convey nuanced findings in the paper is the comparison of GPT-4-generated feedback to human-written feedback. Although the surface-level similarity and helpfulness are similar, my guess is that GPT-4 generated feedback lacks the variability that we’d normally observe in human-written feedback, which can easily affect these models’ performance (e.g., Lee et al., 2023). The limitation section in the appendix mentions coverage of GPT-4 generated feedback, which is relevant but different from this point.
* Similarly, although I generally agree that this is a step towards including more real-world-like elements compared to evaluating LLMs without tool use and natural language feedback, I wouldn’t say that the current setup is strictly “more aligned with real-world applications” than prior work as there is a clear trade-off due to simulating human-written feedback with LLMs and potential drawbacks with the simulation (e.g., Koo et al. (2023), Rajani et al. (2023)).
* In finding #3, could the size of a model be the main confounder? Is the finding only applicable to multi-turn settings or also in single-turn settings?

---

> ### Author Response · Authors · 2023-11-19
>
> We thank the reviewer for their detailed and thoughtful responses, and we are glad that the reviewer found our paper important, sound, and easy to follow.
>
> > …it wasn’t clear to me which ones are new findings vs. confirmation of existing findings……Situating the findings in the existing literature will give this work more validity and help readers understand the contributions of the paper…  As some of the prior work covers a range of diverse tasks/measures multi-turn interaction capabilities of LLMs/simulates human feedback with LLMs, the main difference of this work is (presumably) accounting for the use of tools and user feedback on top of all of these elements… why it’s important to look at these elements altogether (as there are prior work looking at the tool use and NL feedback) and what we expect to see?… I wouldn’t say that the current setup is strictly “more aligned with real-world applications” than prior work as there is a clear trade-off due to simulating human-written feedback with LLMs and potential drawbacks with the simulation (e.g., Koo et al. (2023), Rajani et al. (2023)).
>
> We acknowledge the contributions and findings from existing literature on tool-augmented language models (e.g., ToolLLM) and human-AI interactions (e.g., HALIE). Different from existing work that independently contributed to the understanding of a particular element (e.g., tools, multi-turn, human feedback), to the best of our knowledge, MINT is the first to put everything together.
>
> MINT, by unifying the evaluation dataset for both the tool and language feedback aspect, allows us to examine the interplay between these two elements, i.e., the trade-off between tool performance and the model’s ability to leverage language feedback, which is common in real-world applications (e.g., chatGPT with plugins). We actually can observe such a trade-off in MINT’s evaluation of the Lemur model [1]: we find that their incorporation of reasoning (e.g., OpenOrca) task-specific single-turn data in SIFT improved the model’s reasoning performance (tool-only) without language feedback. However, training on such single-turn data sacrifices their ability to leverage language feedback (e.g., only +0.9 improvement when language feedback is given in Table 3). We will include more discussion on relevant literature and better situate our findings to make it clearer.
>
> [1] Lemur: Harmonizing Natural Language and Code for Language Agents
>
> > … Although the surface-level similarity and helpfulness are similar, my guess is that GPT-4 generated feedback lacks the variability that we’d normally observe in human-written feedback, which can easily affect these models’ performance (e.g., Lee et al., 2023)...
>
> We acknowledge that although it has been more and more widely accepted to use GPT-4 to provide feedback, this approach has the limitations that the reviewer points out (i.e., they might not capture the variability that might present in real human feedback). Possible solutions include explicitly prompting GPT-4 to play various personas, which would complicate the evaluation and out of the scope of MINT but is nonetheless worth exploring. We conjecture that even though our setting (with less variety) does not perfectly reflect the models' performance with real human feedback, it highly correlates with it and provides a good starting point for future research that studies the difference between real-world human feedback and GPT-4 simulated feedback. We will include more discussion and clarifications about this in our next revision.
>
> > In finding #3, could the size of a model be the main confounder? Is the finding only applicable to multi-turn settings or also to single-turn settings?
>
> This is a great question! As discussed in section 3.2, size does play a role in the multi-turn performance: multi-turn tool-augment interaction performance of LLM with the same training data improves with scale. However, since commercial models did not release their actual model size, it is hard to say whether their good performance in multi-turn interaction comes from model scaling versus better data. We believe that finding #3 mainly applies to multi-turn settings since we did not comprehensively test LLM’s single-turn performance with well-established benchmarks (e.g., MMLU, GSM8K).

---

> > ### Comment · Reviewer_hgTu · 2023-11-22
> >
> > Thank you for the response! My questions have been all answered by the authors.

---

### Official Review · Reviewer_vnHy · 2023-10-31

**Soundness:** 4 excellent
**Presentation:** 4 excellent
**Contribution:** 4 excellent
**Rating:** 8
**Confidence:** 5

**Summary:**

The paper investigates the capabilities and limitations of large language models (LLMs) when solving tasks with multi-turn interactions by using tools and leveraging natural language feedback. The main contributions of the paper can be summarized as follows:
(1) The authors propose a structured evaluation framework where LLMs can leverage tools via executing Python code and receive users' natural language feedback;
(2) The authors have conducted abundant experiments, analyzing 20 open- and closed-source LLMs over 8 datasets.
(3) From the experiments, the authors have made some interesting observations, like RLHF may hurt multi-turn interaction performances in problem solving.

**Strengths:**

- The authors introduce MINT, a novel benchmark and framework, that evaluates LLMs' capabilities to solve tasks with multi-turn interactions with tool usage and natural language feedback.
- The authors conduct abundant experiments, covering 20 open- and closed-source LLMs and 8 datasets from different tasks.
- The authors have made some interesting statements, including (1) better single-turn performance does not guarantee multi-term performance; (2) RLHF may hurt the multi-turn capabilities. Corresponding experimental results and analysis are included to support the claims.
- The paper is well-written and is easy-to-follow. The visualizations are also clear and can help readers better understand the framework and model performance comparisons.

**Weaknesses:**

Overall, this is a very thorough and comprehensive benchmark paper. I have only several concerns:
- The authors claim that the SIFT can benefit models' capabilities of tool-augmented task-solving in multi-turn interaction (section 3.2), while the authors also claim that the SIFT can hurt models' ability to leverage feedback (Section 3.3). Why do these two claims seem to contradict each other?
- I am also very curious about the trade-off between tool-use capabilities and abilities to leverage human feedback. SIFT on general domain multi-turn interaction data (like ShareGPT, in section 3.2) can bring performance gain in problem-solving. I think this might be because of the multi-turn feature that caters to the final application. I also think if the model can SIFT on some task-specific data (if exists), the performance can also improve. I am very curious about which is more important for SIFT.

**Questions:**

See weaknesses.

---

> ### Author Response · Authors · 2023-11-19
>
> We thank the reviewer for their detailed and thoughtful responses, and we are glad that the reviewer found our paper novel, important, and easy to follow.
>
> > The authors claim that the SIFT can benefit models' capabilities of tool-augmented task-solving in multi-turn interaction (section 3.2), while the authors also claim that the SIFT can hurt models' ability to leverage feedback (Section 3.3). Why do these two claims seem to contradict each other?
>
> We thank the reviewer for pointing out this confusion! In summary, we find that SIFT trained on **multi-turn** conversations (e.g., Vicuna, Lemur was trained on ShareGPT) have improved tool-use (Section 3.2) and leveraging feedback capabilities (Section 3.3); While for other models (e.g., LLaMA, CodeLLaMA), SIFT can hurt their capabilities. We will make this part clearer in the revision.
>
> > I am also very curious about the trade-off between tool-use capabilities and abilities to leverage human feedback. SIFT on general domain multi-turn interaction data (like ShareGPT, in section 3.2) can bring performance gain in problem-solving. I think this might be because of the multi-turn feature that caters to the final application. I also think if the model can SIFT on some task-specific data (if exists), the performance can also improve. I am very curious about which is more important for SIFT.
>
> We agree that it is an interesting and important trade-off regarding the use of general-domain multi-turn conversation (e.g., ShareGPT) versus task-specific data in improving models with SIFT. In MINT’s evaluation of the Lemur model [1], we find that SIFT on task-specific data (e.g., OpenOrca for reasoning) significantly improved the model’s ability to perform reasoning *without* language feedback; However, the task-specific dataset OpenOrca only contains single-turn data, which we hypothesize could make the trained model more “stubborn,” i.e., sacrificing their ability to leverage language feedback (e.g., only +0.9 improvement in Table 3 when given language feedback).
>
> If our ultimate goal is to improve the model’s single-turn performance (single question-answer pair), we think training on task-specific data might be sufficient. However, If the ultimate goal is to build a helpful interactive assistant that is not only good at providing good answers in the first turn, but can also collaborate with users in multi-turn settings to correct its previous answers (e.g., an almost perfect piece code that only needs a small tweak to work), balancing the model’s ability of direct problem-solving AND leveraging language feedback in a collaborative setting would be essential. We believe a more rigorous study could be done as a follow-up to further analyze each SIFT component’s effect on the model’s different capabilities.
>
> [1] Lemur: Harmonizing Natural Language and Code for Language Agents

---

> > ### Comment · Reviewer_vnHy · 2023-11-20
> > **Official Comment by Reviewer vnHy**
> >
> > Thank you for your efforts during the rebuttal stage. My concerns have been resolved. I will maintain my positive score and vote for acceptance.

---

### Official Review · Reviewer_VNEC · 2023-11-01

**Soundness:** 3 good
**Presentation:** 3 good
**Contribution:** 3 good
**Rating:** 6
**Confidence:** 4

**Summary:**

This paper introduced MINT, a benchmark for evaluating LLMs when they can interact with external tools and natural language feedback. In the benchmark environments, "tools" are considered as functions that can be called by executing the LLM-generated Python code, and "natural language feedback" is simulated using GPT-4. The benchmark covers tasks such as math reasoning and code generation, where only instances that require multiple turns of interaction are retained. In experiments, a comprehensive set of open/closed-source LLMs are evaluated on MINT.

**Strengths:**

1. LLM evaluation in an interactive environment is an important and novel topic.
2. Experiments on a comprehensive set of LLMs under different settings (base vs. SIFT vs. RLHF, open-source vs. closed-source) were conducted, resulting in several interesting observations (e.g., RLHF may hurt LLM-tool interaction while SIFT helps).
3. The paper is easy to follow.

**Weaknesses:**

My major concerns lie in the natural language feedback simulation setup:
1. The example in Figure 1 seems to assume a "collaborative work" interaction between the LLM and the user, as opposed to the more common case where a user seeks assistance from the LLM; in the latter case, it is not practical to assume such detailed feedback from the user because the user does not know how to solve the problem (otherwise they have no need to seek help from the LLM). In some sense, the current LLM-user interaction has made the user a "teacher" who instructs the LLM to correct its mistakes.

    In fact, user feedback has been widely studied in code generation/semantic parsing tasks (references below, which are missing in related work discussion). Prior work has particularly focused on collecting or simulating natural language feedback that is close to "what a prospective, help-seeking user in practice could give". While MINT's feedback interaction is still valuable (as it assesses how well an LLM can incorporate the feedback, no matter if it is realistic or not), I'd like to note that its kind of feedback may not reflect the realistic case, and this should be clarified in the paper.

    References:
    - Ziyu Yao, Yu Su, Huan Sun, and Wen-tau Yih. 2019. Model-based Interactive Semantic Parsing: A Unified Framework and A Text-to-SQL Case Study. In Proceedings of the 2019 Conference on Empirical Methods in Natural Language Processing and the 9th International Joint Conference on Natural Language Processing (EMNLP-IJCNLP), pages 5447–5458, Hong Kong, China. Association for Computational Linguistics.
    - Ahmed Elgohary, Saghar Hosseini, and Ahmed Hassan Awadallah. 2020. Speak to your Parser: Interactive Text-to-SQL with Natural Language Feedback. In Proceedings of the 58th Annual Meeting of the Association for Computational Linguistics, pages 2065–2077, Online. Association for Computational Linguistics.
    - Hao Yan, Saurabh Srivastava, Yintao Tai, Sida I. Wang, Wen-tau Yih, and Ziyu Yao. 2023. Learning to Simulate Natural Language Feedback for Interactive Semantic Parsing. In Proceedings of the 61st Annual Meeting of the Association for Computational Linguistics (Volume 1: Long Papers), pages 3149–3170, Toronto, Canada. Association for Computational Linguistics.

2. The human evaluation of GPT-4 generated feedback does not look at the discrepancy mentioned above either.

3. As indicated in footnote 4, evaluating one LLM using MINT costs around 100 USD, due to the use of GPT-4 for feedback simulation. This cost can be an issue for iteratively improving (and then evaluating) an LLM.

Other than the feedback simulation, another comment on the tool interaction:

4. The current tool use has been limited to Python function calls. It offers better reproducibility but also loses the benchmark scope.

**Questions:**

1. Can the authors respond to my comment on the feedback simulation discrepancy?
2. Can the authors share thoughts on reducing the benchmark cost?
3. Can MINT be integrated with AgentBench (Liu et al., 2023) or InterCode (Yang et al., 2023), so that a larger task scope could be facilitated with both tool and user interaction?

---

> ### Author Response · Authors · 2023-11-19
>
> We thank the reviewer for their detailed responses, and we are glad that the reviewer found our paper novel, important, and easy to follow.
>
> >... Figure 1 … assume a "collaborative work" interaction between the LLM and the user, as opposed to the more common case where a user seeks assistance from the LLM ... not practical to assume such detailed feedback from the user because the user does not know how to solve the problem... While MINT's feedback interaction is still valuable …its kind of feedback may not reflect the realistic case, and this should be clarified in the paper… The human evaluation of GPT-4 generated feedback does not look at the discrepancy …
>
> We thank the reviewer for suggesting relevant work, which we will include in our next revision. For ease of evaluation, in MINT, we disentangle a user’s “assistance-seeking” and “collaborative” feedback-providing behavior.
>
> The “**assistance-seeking**” (not depicted in figure 1) corresponds to (1) the initial user-provided instruction to LLM and (2) templated user feedback LLM received (e.g., Your answer is wrong) when proposing a solution that users disagree with (Lazy User-LLM Interaction section on Page 3, Table 2). As a side note, we think this is similar to real-world user interaction with LLM: even when the user does not know how to solve the problem, it is still possible for the user to provide feedback. For example, I might not know how to write code to solve a particular problem from scratch, so I ask LLM to help, but I might be able to understand and judge the LLM-produced code (and its intermediate code execution result) to see if it fits my expectations and can response with minimal feedback (e.g., "this is not what i want" - similar to the templated feedback). MINT focuses on evaluating LLM on these types of “hard to solve directly, but easy to verify” problems (Table 1).
>
> For the latter “**feedback-providing**” behavior (a more collaborative setting), we use GPT-4 to simulate feedback to mimic the “collaborative” feedback aspect of the user, which is the dotted box shown in Figure 1 with results in Table 3. As described in Section 2.1, we separately evaluated these two as “*LLM-Tool interaction with Lazy User-LLM Interaction*” and “*Informative User-LLM interaction with Language Feedback*.” We will use the suggested wording to explicitly distinguish and clarifies these type of interactions in the revision.
>
> MINT leverages GPT-4 (a stronger / larger model) to simulate human language feedback, which is reproducible and costs 14x less (footnote 4) compared to hiring real humans. We do agree that GPT-4 simulated feedback can be different compared to real-world human feedback, which is evidenced in our human evaluation in Table 5, where we found most human feedback is comparatively less helpful than GPT-4 feedback. Still, such GPT-4 simulated feedback can serve as an initial step to help us gauge LLM’s ability to incorporate feedback (i.e., can they incorporate feedback that’s already very detailed and helpful?), which can provide future research that studies the difference between real-world human feedback and GPT-4 simulated feedback a good starting point. We will include clarifications in our next revision to make this more clear.
>
> > As indicated in footnote 4, evaluating one LLM using MINT costs around 100 USD, due to the use of GPT-4... can be an issue for iteratively improving (and then evaluating) an LLM... thoughts on reducing the benchmark cost?
>
> MINT, similar to the widely used MT-Bench [1],  chose to use GPT-4 due to its effectiveness and reproducibility. We are aware that such a cost might become a roadblock for iterative improvement, and therefore, we made significant efforts to keep MINT’s evaluation dataset compact and representative to reduce the GPT-4 feedback cost down to ~100 USD (98% less compared to the original size of the datasets). Furthermore, Human evaluation costs on a similar quantity of examples are likely to cost 14x more (footnote 4), which already makes GPT-4 feedback simulation more cost-effective.
>
> In our design, we deliberately made the feedback provider replaceable (Section 3.4), which allows potential future cost-saving by replacing GPT-4 with a cheaper model that specializes in providing feedback (e.g., [2]).
>
> [1] Judging LLM-as-a-Judge with MT-Bench and Chatbot Arena
>
> [2] Shepherd: A Critic for Language Model Generation
>
> > The current tool use has been limited to Python function calls. It offers better reproducibility but also loses the benchmark scope.
>
> Please refer to the general response.
>
> > Can MINT be integrated with AgentBench (Liu et al., 2023) or InterCode (Yang et al., 2023), so that a larger task scope could be facilitated with both tool and user interaction?
>
> Yes! We do plan to continuously integrate additional tasks (e.g., AgentBench, InterCode) into MINT to further expand its scope in the future (e.g., a MINT 2.0 version) as MINT’s evaluation framework is general and task-agnostic.

---

> > ### Comment · Reviewer_VNEC · 2023-11-22
> > **Thank you for your response!**
> >
> > Thanks to the authors for answering my questions!
> >
> > > For example, I might not know how to write code to solve a particular problem from scratch, so I ask LLM to help, but I might be able to understand and judge the LLM-produced code (and its intermediate code execution result) to see if it fits my expectations and can response with minimal feedback (e.g., "this is not what i want" - similar to the templated feedback). MINT focuses on evaluating LLM on these types of “hard to solve directly, but easy to verify” problems (Table 1).
> >
> > I think the authors' response is a bit misleading. Clearly the feedback shown in Fig 1 has went beyond "this is not what i want". The demonstrated feedback, e.g., "You should have used ... Then you can express...", is procedural and detailed, which may have assumed that the user knows the exact step-by-step answer. So again it looks more like a user teaching LLMs. If a user knows the problem with so many details, why don't they directly edit the code, which could even give a better correction guarantee than natural language feedback? A fundamental question here is, who will be the prospective user under this framework's formulation? People who cannot solve math problems completely but actually knows the procedures, can read code but cannot directly edit code?
> >
> > I understand that this work is mainly to evaluate the feedback incorporation of LLMs, which is very exciting to me and I'm still holding a positive opinion on it. However, I hope the authors could have a more in-depth discussion on the envisioned user-LLM interaction scenario and how it connects to real-world applications/user groups (or not), and acknowledge the effort of prior work in bridging these important gaps. A good example is the work of Elgohary et al., which introduced natural language explanation of SQL code such that the interface can benefit users without SQL programming expertise (i.e., who cannot read SQL code nor edit it).
> >
> > Regarding the tool use, I agree that it is often converted to code gen. Thanks for the reminder of F.1.

---

> > > ### Author Response · Authors · 2023-11-23
> > >
> > > Thanks for the follow-up!
> > >
> > > > Clearly the feedback shown in Fig 1 has went beyond "this is not what i want". The demonstrated feedback, e.g., "You should have used ... Then you can express...", is procedural and detailed, which may have assumed that the user knows the exact step-by-step answer.
> > >
> > > Sorry for the confusion! We would like to clarify that “**assistance-seeking**” behavior like “this is not what i want” is not depicted in figure 1, but rather described in *Lazy User-LLM Interaction section* on Page 3. We consider these type of simple feedback the minimal feedback you can expect from a user. We will try to update Figure 1 in our next revision to better reflect this type of user behavior. Here is our description of Lazy User-LLM interaction in Section 2.1 of the paper:
> > >
> > > Lazy User-LLM Interaction. We consider the scenario where a user provides an initial instruction and **makes minimal efforts to guide the LLM towards the final solution**. This will serve as a baseline for subsequent evaluations of LLM’s ability in tool-augmented task-solving and leveraging natural language feedback.
> > >
> > > > If a user knows the problem with so many details, why don't they directly edit the code, which could even give a better correction guarantee than natural language feedback? A fundamental question here is, who will be the prospective user under this framework's formulation? People who cannot solve math problems completely but actually knows the procedures, can read code but cannot directly edit code?
> > >
> > > That is a great question! We do think there are a lot of cases where this could happen:
> > >
> > > For example, a novice user that do not know coding might want to build a beautiful visualizer for their tabular data. After the first turn interaction with LLM, they obtained the code (but they don’t fully understand). However, they might be able to directly start the visualizer to see if the visualization is something they want (easy for novice user). If not, it is easy for them to provide high-level language feedback to point out issues (e.g., “1st column and 2nd column are aligned”), and then a LLM that can leverage language feedback will be able to take that feedback and improve the code.
> > >
> > > In another example, a professional user might want the help from LLM to write a deep learning training script. The user looked at the code generated in first interaction turn, everything is perfect except that the user would like to point out some part of the implementation is too slow (e.g., nested for-loop) and can be make much faster (e.g., using vectorization). In this case, the user can definitely copy the code to their IDE, manually edit the code and debug it to make it work – But it will boost the productivity of the user much more if they can simply provide language feedback like “The nested for-loop is slow, give me something faster”, or a more detailed version like “The nested for-loop is slow, consider using vectorization,” and the LLM will handle it from there and improve the code.
> > >
> > > Therefore, in MINT, we are considering language feedback in two extreme: (1) minimal - (Lazy User-LLM interaction, e.g., “This is not I want”, described in Section 2.1); and (2) detailed - (Informative User-LLM Interaction with Language Feedback, simulated by GPT-4, e.g., “The nested for-loop is slow, consider using vectorization.”). We expect language feedback that come from actual human users (both expert and novice) could lies between (1) and (2). We will provide more in-depth discussion and better situated our work with existing literatures in the next revision.

---

### Author Response · Authors · 2023-11-19

We would like to express our gratitude to the reviewers for their time and effort in reviewing our paper, as well as their constructive comments and suggestions.

In this general response, we hope to address a concern raised by multiple reviewers (VNEC and Nxjw) regarding the choice of using Python as an interface for tool calls.

> [VNEC] The current tool use has been limited to Python function calls. It offers better reproducibility but also loses the benchmark scope.

> [Nxjw] It seems that the only tool used is the python execution tool. Compared to other papers leveraging tools (toolformer, ReAct), the claim of utilizing tools through turns would improve model performance is not very convincing, especially for tasks that may not require code execution (e.g., HotpotQA)... MINT is limited to multi-turn interactions using the specific code tools….

We beg to differ. On the contrary, we consider using Python as an interface for tools being a superset of current tool use paradigms that naturally encapsulate the Toolformer / ReACT framework the reviewers raised.

Human-provided tools to LLM, which is prevalent in existing LLM’s tool-use setting, can be simply cast into a custom Python function. For example, on HotpotQA tasks that the reviewer Nxjw mentioned, we provided the model with a Wikipedia search function (Section F.1 in Appendix) that allows retrieval to be achieved through code execution. As our best effort, we provided task-related tools that are sufficient to solve these tasks if the LLM can use them correctly.

We argue that tool calls, in essence, are code execution. ToolFormer and ReAct represent each tool call with a different text format (e.g., “Action:”), which will be later parsed into programming language function calls for execution in order to obtain observations. In this work, we use Python code as a format to unify tool calls for all tools we are evaluating due to its flexibility.

In addition, the flexibility of Python code allows LLM to access a wide array of Python built-in and third-party tools/packages that LLM was not explicitly told to use. For the latter case, since most LLMs have been extensively trained on code data, they know how to use Python built-ins like sort, min, and max, perform arithmetic operations, and use advanced third-party libraries (e.g., sympy) without the user explicitly providing such tools. We believe using Python as an interface for tool calls can demonstrate the model's potential beyond the scope of our provided tools, aligning with our goal of evaluating the model's ability to use tools in diverse contexts.

---

### Meta-Review · Area_Chair_Q5sC · 2023-12-05

**Metareview:**

This paper introduced MINT, a benchmark to evaluate LLMs in solving problems via multi-turn interactions with tools and natural language feedback. MINT comes with 586 problems derived from existing “single-turn” datasets from a variety of domains, including coding, reasoning, decision making and question answering. MINT uses Python programming language as a general-purpose interface to specify the semantics of tools and the Python interpreter as the sandbox execution environment. Execution results are used by GPT-4 as a user simulator to generate natural language feedback. Extensive evaluation is conducted across 20 open- and closed-source LLMs.

**Strengths**

* This is a well-written paper (vnHy, VNEC, hgTu) that aims to measure progress on the important and timely topic of multi-turn interactions for tool use (hgTu, VNEC), and the resulting MINT benchmark would be a nice contribution to the community (vnHy, Nxjw).

* All reviewers found that this paper presents comprehensive experiments over 20 LLMs on 8 task varieties, leading to interesting insights. Overall, the results suggest the importance of improving multi-turn skills of LLMs, especially during instruction tuning and RLHF.

* Using GPT-4 to simulate user-issued natural language feedback is a scalable and cost-effective approach (hgTu), although it's less clear if such simulated feedback matches with real-world scenarios (see weaknesses-2 below).

**Weaknesses**

* The reviewers (VNEC, Nxjw) were initially concerned with the generality of using Python as a unified interface for tool invocation. This issue was resolved during the rebuttal phase, and I suggest the authors include an explanation in the final version.

* Reviewer VNEC raised a valid concern that there is a discrepancy between the GPT-simulated feedback and “what a prospective, help-seeking user in practice could give” in real-world scenarios. While the author’s response attempted to justify that MINT covers both “assistance-seeking” feedback with minimal information and more collaborative feedback, it’s still unclear if the feedback from real-world users under these two categories follow a similar style as those simulated ones. It would be good for the authors to include the discussions in their next draft.

* Missing references to existing research on incorporating user feedbacks in semantic parsing (VNEC), or prompting-based self-improvement methods (Nxjw)

* Minor issues with the presentation in terms of the scope of contributions and disentangling new and existing findings (hgTu), as well as when supervised fine-tuning may hurt tool-use performance (vnHy)

Overall, the paper is well-written with comprehensive evaluations, and there are no significant issues with the benchmark construction and evaluation methodology. The decision is *“Accept”*.

**Justification For Why Not Higher Score:**

* This is mostly a benchmark paper without inventing new methods.

* There is an open-issue regarding the mis-match between the simulated and real-world user feedback to LLMs in tool-use applications, which may affect the perceived excitement from audience.

**Justification For Why Not Lower Score:**

* The paper is well-written with comprehensive evaluations and the associated MINT benchmark is an important contribution to the community.

* Three out of four reviewers would love to see this paper being accepted.

---

### Decision · Program_Chairs · 2024-01-16

Accept (poster)